# Ill Fares the Land: Confronting Unsustainability in the U.K. Food System through Political Agroecology and Degrowth

Mark Tilzey 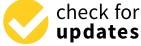

Centre for Agroecology, Water and Resilience, Coventry University, Coventry CV8 3LG, UK;
ab7997@coventry.ac.uk

**Abstract:** The U.K. food system exhibits strong unsustainability indicators across multiple dimensions, both in terms of food and nutritional insecurity and in terms of adverse climate change, biodiversity, and physical resource impacts. These indices of an unsustainable and inequitable social metabolism are the result of capitalist agriculture and society in general and, more specifically, of neoliberal and austerity policies adopted with vigour since the global financial crisis. The causal, capitalistic, and, latterly, more neoliberal bases of the U.K. food system are delineated in the first section of the paper. These bases are then detailed in terms of their impacts in exacerbating climate change, biodiversity (and resource) decline and loss, and food and nutritional insecurity. The political narratives and policy frameworks available to dissemble, mitigate, or, more rarely, to address (resolve) these impacts are then delineated. It is argued that the only policy framework available that strongly integrates food security (social equity) with ecological sustainability is political agroecology and an accompanying degrowth strategy. The final section of the paper details what political agroecology and degrowth might entail for the U.K. food system.

**Keywords:** U.K. food system; capitalism; neoliberalism; climate change; biodiversity; food insecurity; political agroecology; food sovereignty; degrowth

## 1. Introduction

A recent and authoritative assessment of the U.K. food system [1] concluded that it currently fails the test of sustainability on multiple criteria—food security, dietary quality and equality, greenhouse gas (GHG) emissions, biodiversity and soil conservation, and so on. The author (Lang) makes an incontrovertible case for the inherent and growing fragility of the U.K. food system across these dimensions, especially, perhaps, in relation to the historical and current reliance of the system upon the U.K.'s ability to buy 'food from nowhere' according to the principle of 'cheapness', without asking necessary questions concerning the ecological and social sustainability of the food so produced. Lang's detailing of this fragility is comprehensive; his analysis and critique are undertaken from what he terms a 'multi-criteria' or what might be called a 'systems-theoretical', perspective. This has the advantage, in contrast to neoclassical economic ('hegemonic') theory, of bringing into play all dimensions of the food system, placing on an equal footing the 'economic', 'political', 'socio-cultural', and 'ecological' variables involved. While this approach is essential as a starting point, it becomes less convincing the more questions are asked concerning causality underlying the dynamics of the system. In other words, the question of 'structural causality', or the principal causal driver(s), underpinning and propelling the system seems to be lacking. This is unfortunate because structural causality, following the principles of critical realism, enables us to make sense of the way in which the above variables interact in a causal hierarchy. Thus, critical realism synthesises multiple determinations, identifies the underlying real mechanisms, and connects them to actual and empirical aspects of the *explanandum*, the phenomenon to be explained [2–4]. This missed opportunity on Lang's part to drill down analytically into structural causality has an important impact on his

normative proposals for rendering the U.K. system more food-secure and resilient. Notably absent amongst these is a critique of capitalism, surely the *primum mobile* of a system that is predicated on market hegemony, exponential growth and consumption, social inequality, and the externalisation of ecological and social costs—all features central to the contradictions of the *capitalist* U.K. food system. Lang, however, appears to assert that the search for a more food secure U.K. should be deployed as a test of the strength of British capitalism [1], as if the two aims were somehow entirely compatible and natural bedfellows.

This paper, while covering territory similar to Lang's book [1], develops a rather different line of analysis, with concomitantly rather different normative conclusions. This is not to suggest that the two analyses and their respective normative recommendations are incompatible but rather to aver that the political and ecological transformation of the U.K. food system required to render it sustainable is likely to be rather deeper and more comprehensive than Lang allows. Given the deep ecological and social contradictions of the U.K. food system that we will delineate in this paper (contradictions integral to the unsustainability of the U.K. capitalist economy as a whole), it appears appropriate, therefore, to build on the more radical analyses and normative proposals for transformative change set out in a series of recent publications addressing political ecology, agroecology, food sovereignty, and degrowth [4–7]. In line with these radical analyses, a recent paper by Guerrero Lara et al. [8] identifies and synthesises a useful research agenda for critical agrarian studies in relation to the degrowth problematic, and we propose, in the present paper, to engage with this agenda in relation to the U.K. food system (while noting that significant elements of this agenda have in fact been parsed in the aforementioned publications, albeit not necessarily in relation specifically to 'degrowth'). These authors identify four areas in need of further research and development in relation to agri-food studies and degrowth: degrowth conceptualisations, theorisation of transformations towards sustainability, the political economy of degrowth agri-food systems, and rurality and degrowth.

Concerning the first area, degrowth conceptualisations, Guerrero Lara et al. point to the relative analytical neglect of the ecological conditions and the energy/material throughput of proposed alternative agri-food systems by comparison to the more common focus on the social principles of degrowth. Research that identifies and quantifies possible changes in social metabolism and nutrition, they suggest, can serve as a 'reality check' in relation to assertions regarding the potential of alternative agri-food models to reduce energetic/material throughput (ecological sustainability) whilst sustaining or enhancing human nutritional standards and social well-being/equity (social sustainability). More specifically, they ask, 'what is the social metabolic space of possibilities for the reduction of material and energy throughput in agri-food initiatives from food production to consumption to make them "thermodynamically efficient" rather than striving for more economically efficient modes of consumption and production' (p. 1583). It is also necessary, of course, to gauge the biophysical indices of the current (capitalistic) social metabolism in order to assess their degree of (un)sustainability and, therefore, the level of transformation required to render the social metabolism 'thermodynamically efficient'. Building on previously published work [4,9,10], these tasks we undertake in the present paper in sections addressing the climate change, overseas land footprint, and biodiversity impacts of the U.K. food system and the outlining of a policy framework for agroecology, food sovereignty, and degrowth.

Within this first area (degrowth conceptualisations), Guerrero Lara et al. also point to the need for degrowth research in agri-food systems to engage more with literature addressing policy and social movements. They suggest that few studies have investigated the role of policies such as those governing trade and agriculture as factors influencing the degrowth transformation of agri-food systems. While this unfortunately neglects significant studies that have addressed the constraints on transformations to sustainable food systems (implying or specifying degrowth) imposed by varying configurations of capitalism [11–15], the role of policy and its relationship to social movements (that is, the relationship between 'systemic' and 'anti-systemic' agents) in the dynamics of transformative change are discussed, in varying degrees, in all sections of the present paper. Work specifically addressing

the relationship between 'systemic' ('hegemonic' and 'sub-hegemonic') and 'anti-systemic' ('alter-hegemonic' and 'counter-hegemonic') agency in relation to agri-food systems has also been presented in detail elsewhere [4,10,15,16].

In respect of the second area identified by Guerrero Lara et al., the theorisation of transformations towards sustainability, these authors assert, correctly, that research has commonly lacked a consideration of capitalism, has usually related to 'Western' countries, and, by the same token, has been of limited applicability to 'non-Western' societies. While this criticism is certainly well-founded as a generality, it is important to note work that has attempted to address these lacunae [4,10,12,15–18] since this forms an essential backdrop to the current paper. Guerrero Lara et al. highlight three themes within this area that would, they maintain, benefit from greater research focus: learning from critical perspectives within sustainability transformation scholarship, investigating the multiplicity of change agents beyond grassroots initiatives, and bringing in the more-than-human dimension.

Inter alia, they suggest the following:

- Research addressing peasant/indigenous movements and decoloniality should move centre stage—here, the following publications should be noted [4,9,10,15,16,18,19], and the present paper will build on this work by specifying the 'ecological imperialism' of the U.K. food system;
- Research should ask what role multiple agents of change might play in a degrowth transformation and how their political agendas might intersect or conflict—here, complementing prior work addressing political (class) positionalities and discourses [4,10–12,15,16], the present paper will delineate these positionalities and discourses and argue that dominant class interests in the U.K., in upholding capitalism in various forms, impose strong constraints on a degrowth agenda;
- More research is needed to identify, critique, and theorise the roles that state and non-state and systemic or anti-systemic actors may have in promoting or inhibiting degrowth transformation of agri-food systems—here, again, Guerrero Lara et al. appear to have overlooked work in precisely this area [4,10,15–17], and in the present paper, we build on this theorisation of the state in relation to systemic and anti-systemic class interests to explore the dynamics of the U.K. food system;
- Finally, in bringing in the more-than-human dimension, this paper will continue the theme developed elsewhere [4,9,10,15,20] that ecological sustainability should be a fundamental desideratum of an enduring and stable social metabolism. Moreover, the paper will address explicitly the relationship between the U.K. food system and biodiversity conservation.

Turning to the third area highlighted by Guerrero Lara et al., the political economy of degrowth in agri-food systems by recentring capitalism, these authors rightly argue that the transformation to a degrowth society cannot be envisaged without conflict in a growth-dependent capitalist system. Inter alia, the authors point importantly to the need for more exploration of how the mechanisms of capitalist institutions impede the success of degrowth agri-food initiatives, how they may be contested, and what alternatives to capitalism may be sought. For instance, farmers confront *structural* constraints imposed by a capitalist regime of private landownership (absolute property rights), pushing them to cultivate in a productivist manner and largely prohibiting agroecological initiatives towards degrowth through lack of access to land. The authors ask, crucially, 'in a society predicated on private property ownership, what elements need to be unmade as part of a degrowth transformation to ensure the decommodification of land and prioritise the use value of land over its exchange value? How can the degrowth movement pursue large-scale land decommodification?' (p. 1588). In raising the issue of structural politico-economic constraints, the authors point to the need to situate agri-food degrowth initiatives within the wider context of capitalist food regimes and within the context of close intersectoral linkages within a state-defined economy as a whole. The present paper addresses these concerns throughout but especially in sections addressing the dynamics of the U.K. food system, contested policy discourses, and a policy framework for agroecology, food sovereignty,

and degrowth. Again, these sections represent an outgrowth of work published elsewhere addressing these themes [4,10,12,15,17,19].

The last area highlighted as requiring further research by Guerrero Lara et al. is that of rurality and degrowth. They point out that most agri-food degrowth research in the global North is concentrated in urban and peri-urban areas—this is due to the concentration here of the greatest economic precarity and consequently of oppositional movements. The countryside in the global North has largely lost its peasantry and is the home of relatively prosperous, conservative, and property-owning farmers, constituting a rather uncongenial environment for oppositional movements [4,10,15,16,19,21]. As Guerrero Lara et al. ask, 'how can degrowth speak to large-scale [or indeed to any market-dependent family] farmers who have been formed and shaped by the capitalist economy's ruthless paradigm of continuous growth?' (p. 1589). Despite widespread antipathy amongst the family farm constituency to ideas associated with agri-food degrowth, many smaller market-dependent farms appear unlikely to survive as economically viable entities beyond the short term, especially in more neoliberally inclined economies such as the U.K. Given that such rising threats to family-farm livelihoods, arising especially from neoliberalisation, appear currently to be resulting in the political embrace of right-wing populism amongst this constituency, it is important to ask, 'how might degrowth help to effectively fight rural marginalisation and decline?' (p. 1589). In the present paper, we attempt to answer this question in the section describing a policy framework for agroecology, food sovereignty, and degrowth whilst acknowledging, in the previous section on contested agrarian discourses, the scale of the material and ideological task in securing alignment to such a politico-ecological transformation.

The present paper is structured as follows. We begin, in the first section of the paper, by delineating the causal, capitalistic, and, latterly, more neoliberal bases of the U.K. food system. In the next three sections, we then detail the empirical indices of unsustainability demonstrated by the U.K. food system, focusing on their impacts in exacerbating climate change and overseas land footprint, biodiversity (and natural resource) decline and loss, and food and nutritional insecurity. We argue that these indices are the result of capitalist agriculture and society in general and, more specifically, of neoliberal and austerity policies adopted enthusiastically by U.K. governments since the global financial crisis and, especially, since 2010. In the fifth section, the principal political narratives and policy frameworks being articulated either to obstruct action, to mitigate, or, more rarely, to address (resolve) these impacts are then delineated. These are defined as 'hegemonic' (neoliberal), 'quasi-hegemonic' (environmental neoliberal), 'sub-hegemonic' ('state–capitalism'), 'alter-hegemonic' (ecocentric), and 'counter-hegemonic' (agroecology-food sovereignty). It is argued that the only political narrative and policy framework available with the capacity to strongly integrate food security (social equity) with ecological sustainability is political agroecology as food sovereignty as part of a programme of degrowth. The final section of the paper outlines what political agroecology and degrowth might entail for U.K. food system.

## 2. The U.K. Food System and Causal Basis of Agrarian Growth and Unsustainability: From National Developmentalism, through 'Embedded' Neoliberalism, to 'Radical' Neoliberalism

### 2.1. National Developmentalism

The current fossil-fuel, agrochemical-based, and fully capitalist[1] configuration of British agriculture broadly achieved its current form during the post-Second World War period. This was a period characterised by Fordism and the development of sectoral articulation between agriculture and industry and social articulation between a fully proletarianized workforce in its role as both producer and consumer, in which Britain, following the severe food insecurity of the war and disruptions to food imports, sought to become self-sufficient in the production of principal food staples. This brought home to Britain the ecological contradictions of productivist agriculture, albeit now intensified through greater dependence of fossil fuels and agrochemicals, that had previously been externalised

onto the spaces of export agriculture abroad during the period that Tilzey [4,17] denotes as the 'Second or Imperial' food regime. A massive acceleration in labour productivity and yields due to fossil fuel-powered mechanisation and agro-chemicalisation, with ecological costs in terms of GHG emissions, soil degradation, water pollution, and severe biodiversity loss effectively externalised [20], was structurally tied to a particular phase of capitalist development that we may term 'national developmentalism' [19,22]. As applied to the agriculture sector, we may refer to this as the 'Third or Political Productivist' food regime [17,20], a state-managed policy framework to which an acceleration of the processes of 'appropriationism' and substitutionism'[2] were pivotal. 'Political productivism', thus embodied in U.K. post-war policy and subsequently in the Common Agricultural Policy (CAP) of the European Union (EU) (to which the U.K. acceded in 1972, only to leave again, as part of 'Brexit', in 2021), was implemented by deploying the instruments of guaranteed prices, investment grants, input subsidies, state regulation of major commodity markets, and their insulation from overseas competition [9,20].

The result was to 'hothouse' agrarian capitalism through a policy framework in which higher net farm income could be secured only by means of productivity and yield increases. This acted as a massive incentive to cut costs through the substitution of machinery for labour, enlarging holdings, and borrowing money for land purchase and capital projects, all dependent centrally on increased fossil fuel and agrochemical consumption. This, in turn, created indebtedness, further reinforcing the imperative to cut costs and increase output [24], enabled by the 'cheap' 'ecological surplus' afforded by fossil carbon extracted primarily (before North Sea oil) from the global South under the aegis of imperially installed autocratic regimes. The appropriation of such 'ecological surplus' (principally through Surplus Extraction 2 (see below)), enabled the fossil-fuel based capitalisation of agriculture, the final elimination of peasant-based production, and the 'secure' employment, as a 'labour aristocracy', of the resulting really subsumed proletariat in urban industry (facilitated also by the racialised super-exploitation of immigrant labour from the periphery as Surplus Extraction 1 (see below))—the result was the full instantiation, under Fordism, of the 'imperial mode of living' [12,25][3].

*2.2. Embedded Neoliberalism*

From the 1980s, however, the regime of 'political productivism' conceded gradually to a more neoliberal (or 'market productivist') regime of accumulation within the CAP, with commodity support measures giving way to direct payments supplemented by discretionary budgets for agri-environmental measures designed to mitigate the more egregious impacts of productivism on the ecosystems of rural areas [9,20]. Tilzey [4,17,26] nominates this new regime the 'Fourth or Neoliberal' food regime, since, like Bonanno and Wolf [27] and Otero [28], he considers the term 'neoliberal' to capture explicitly the central role of the state in re-regulating for and undergirding the strategies of certain fractions of capital, especially those with a transnational orientation. This trend towards neoliberalism, articulated by the EU and supported strongly by the U.K. as a then member state, reflected in no small part the greatly increased influence of transnational and neoliberally inclined class interests in defining and promoting a more globally and market-oriented agricultural policy [11–13]. 'Hegemonic' neoliberalism gradually gained ascendancy vis-à-vis those 'sub-hegemonic' (neo-mercantilist and social protectionist) class constituencies that had formed the bedrock of national developmentalist Fordism [11,12]. This neoliberal strategy was designed to stimulate the further expansion of productivism, now of a more market-oriented kind, and its increased integration into global agri-food circuits of capital. The progressive elimination of 'market distorting' commodity support in favour of (WTO-compatible) direct payments (Pillar 1 of the CAP) was complemented by the creation of Pillar 2 (rural development and agri-environmental monies), designed to afford some measure of continuing support to farmers marginalised in the neoliberalisation process, to provide countryside consumption spaces for the urban populace (while conserving a residual biodiversity and landscape resource), and to supply the new 'health conscious' and

'environmentally aware' middle-class, a 'reflexive' consumer with organic and/or locally produced food commodities. An (asymmetrical) bipolarity in policy became increasingly evident during the 1990s and the new millennium, therefore, between globalising norms of governance for market productivism on the one hand and regionalised or 're-territorialised' norms of governance for 'post-productivist' and 'multifunctional' agri-rural activities on the other. These latter 'ecologised' and 'localised' constituencies we nominate 'alter-hegemonic' interest groups [4,15].

Thus, while changes to Pillar 1 were designed to facilitate the progressive penetration of globalising market relations and international market dependency into EU and British agriculture, Pillar 2 budgets (far smaller than those allocated to Pillar 1), simultaneously, came to be disbursed on a competitive and selective basis and were (and remain) heavily constrained in their ability to counteract the overarching processes of neoliberal restructuring. Budgets for agri-environmental management likewise came to be defined and defended increasingly according to neoclassical 'public goods' criteria (see below), entailing more restrictive forms of subvention in line with WTO 'green box' disciplines and the accompanying requirement to minimise 'market distortion' [4,12,13]. Despite these clear shifts towards neoliberal governance norms, the EU insisted on the retention of significant direct supports for farmers within Pillar 1 of the CAP, the function of which was and remains to act as a WTO-compatible 'hidden subsidy' to Europe's farmers for reasons of political legitimacy, the continuation of which, in the face of global Southern opposition, comprised an important factor in the breakdown of the WTO Doha Development Round of free trade negotiations in 2008 [4]. Such continuing and generous direct subsidy within Pillar 1, together with the very existence of Pillar 2, indicate that the CAP cannot be described as unambiguously neoliberal. Rather, the Polanyian-derived term 'embedded neoliberalism' seems more apposite [12] since, however attenuated the 'agricultural welfare state' [29] might now appear by comparison to its Fordist heyday, the CAP continues to fulfil both the accumulation and legitimacy functions of the 'state'.

*2.3. Radical' Neoliberalism*

Indeed, it was the very retention of such legacies of 'sub-hegemonic', social democratic Fordism that incurred the ire of doctrinaire neoliberals (and their right-wing 'authoritarian populist' allies) in the U.K., especially amongst members of the Conservative Party and in state departments such as the Treasury and DEFRA (the Department of Environment, Food, and Rural Affairs). Harnessing the growing discontent of the British working classes, especially with globalisation and austerity following the financial crisis of 2008, and deploying the EU as a convenient scapegoat for these ills, right-wing 'authoritarian neoliberals' of the Conservative Party contrived to engineer a 'Brexit' departure from Europe under the guise of the national populist slogan 'taking back control', thereby disguising its actual Thatcherite agenda of untrammelled neoliberalism (modified by certain necessary, populist, but minimal concessions to its new proletarian constituency). The Conservative Party and DEFRA were able to portray Brexit as an unprecedented opportunity to address the ecological disbenefits of agricultural productivism (including, by implication, climate change impacts) since, according to neoclassical and neoliberal economic doctrine, these were the outcome not of capitalism but rather of the continuing market interventionism (statism) of the of the EU's CAP. Remove such interventionism, and the 'free play' of market forces would secure that axiom of neoclassical theory, 'optimal allocation of scarce resources'. With this in place, any environmental 'market failures' could then be made good with state subvention for 'public goods'.

The mainstream environmental movement in the U.K. has long held a similar view of the CAP and has, therefore, tended to be beguiled by neoliberal and neoclassical economic arguments for the freer play of 'market forces' as putatively the best means, with the added proviso of 'public goods' payments, to assure environmental sustainability (including climate change mitigation) [9]. The CAP has thus been a relatively easy target for the U.K.'s mainstream environmental conservation movements, precisely because such

critiques sit comfortably with the new neoliberal economic agenda of the Conservative U.K. government and its intention to dismantle the 'distorting' influence of 'sub-hegemonic' market/direct support structures [9]. This agenda is now being enacted post Brexit with the phasing out in the U.K. of inherited CAP supports through Pillar 1 over the period up to 2028 and the intended restriction, thereafter, of state subvention in the agriculture sector to neoliberally configured and WTO-compatible 'public goods' payments. It is this 'hegemonic' discourse of neoliberalism and 'quasi-hegemonic' *environmental* neoliberalism (see below), propounded by the Conservative Party and the principal departments of state (and supported, albeit cautiously, by the main environmental NGOs), that is, in combination, the main determinant of U.K. climate change mitigation, agri-environmental, and food security policy. Below, we examine this discourse, together with other 'sub-hegemonic', 'alter-hegemonic', and 'counter-hegemonic' discourses in relation to agrarian climate change, biodiversity loss mitigation, and food security policy. Before doing so, we lay out the current climate change, biodiversity, and food in/security impacts of the U.K. alimentary system.

### 3. Current Climate Change Impacts and Overseas Land Footprint of the U.K. Food System

*3.1. U.K. Food System Climate Change Impacts and Land Footprint*

In the first study of its kind, Audsley et al. [30] employed a detailed inventory of emissions developed from the life cycle analysis of a wide range of foods and processes comprising three parts: primary production to the regional distribution centre (RDC), from the RDC to consumption (through retail and cooking), and land-use change (LUC). On this basis, they estimated that the supply of food and drink to the U.K. results in a direct emission equivalent of 152 $MtCO_2$ (million tonnes of carbon dioxide equivalent, a figure including other GHGs, notably methane [$CH_4$] and nitrous oxide [$N_2O$]). A further 101 $MtCO_2e$ from LUC, mainly overseas, largely due to deforestation and forest degradation in Latin America and Southeast Asia, is attributable to U.K. food consumption. Total U.K. economy consumption emissions are estimated to be about 748 $MtCO_2e$ (excluding LUC, which, if added, sums to 849 $MtCO_2e$). This means that direct emissions from the U.K. food system comprise about twenty percent of the currently estimated consumption emissions of the U.K. economy, a figure that rises to thirty percent if LUC emissions are added [30,31].

Of direct emissions (excluding LUC), fifty-eight percent arise from animal production and products, which, however, account for only thirty percent of consumer energy intake [30–32]. Overall, about twenty percent of direct U.K. food chain emissions occur outside the U.K. However, if LUC is taken into account, this figure increases dramatically to around fifty percent, meaning that about half of total U.K. food system emissions arises outside the country. Audsley et al. conclude that the direct and indirect (that is, LUC) effect of the supply of the food to the U.K. as a contributor to global land-use change pressures is a significant factor in U.K. consumption emissions. Their study also attributes a large proportion (seventy-five percent) of LUC emissions to ruminant meat production, primarily through the production of soya feed for beef and dairy, and to lesser extent through direct beef exports, sourced overwhelmingly from Latin America on areas formerly characterised by biodiverse, high carbon sequestration biomes.

Primary production, that is, production of food commodities, accounts for fifty-six percent of direct emissions (excluding LUC), with nearly half comprising $N_2O$ from agricultural soils through the application of synthetic fertilisers and $CH_4$ from enteric fermentation from ruminant livestock (primarily cattle and sheep). The source of the other half or so is dominated by $CO_2$ emissions from fossil energy used in the manufacture of agricultural inputs, such as energy use in highly mechanised farming, commodity storage, and some processing. Beyond primary production, energy use in processing, manufacture, transport, retail, and food preparation/cooking accounts for thirty-seven percent of all direct emissions. However, if we again factor in LUC, this contributes, as noted, around 100 $MtCO_2e$, most of which comprises $CO_2$ emissions through direct destruction or degradation of high

carbon sequestering biomes in the global South. Again, this indicates that LUC, located primarily in the highly biodiverse and high carbon sequestering biomes of the global South (Latin America (soya and beef) and Southeast Asia (palm oil)) comprises the single largest contributor to U.K. food system GHG emissions, with emissions from primary production and then RDC to consumption (through retail and cooking) representing the second and third most important sources of U.K. food system GHG emissions [30]. This represents very considerable 'carbon leakage' (the displacement of GHG emissions to states outside those where product consumption occurs) in the U.K. food system.

*3.2. Impacts Increased by Meat- and Dairy-Heavy Diets*

GHG emissions in primary production and in LUC are exacerbated by meat- and dairy-heavy diets characteristic of the 'imperial diet' since meat production is energetically less efficient than producing plant foods directly for human consumption. Moreover, ruminants produce the potent GHG $CH_4$ as a by-product of digestion. GHG emissions are exacerbated still further when ruminants are fed grains produced on the basis of fossil energy-dependent mechanisation and agrochemicals, the latter producing the most potent GHG $N_2O$, some 265 times as virulent as $CO_2$. The worst emissions occur when primary production as specified above is preceded by LUC that entails the destruction or degradation of high carbon sequestering and climate stabilising native vegetation (it is known that trees emit natural aerosols that act as water vapour condensation nuclei, which then both cool the air within and around forests and increase local rainfall, significantly mitigating climate change *in addition* to their role as carbon sequestrators [33]). Sadly, much of the feed grown in the global South and destined to feed ruminants in the U.K. is produced under the 'worst case' scenario described above. In this regard, de Ruiter et al. [34] endeavoured to detail the total agricultural land footprint associated with the U.K. food supply, differentiating between the impacts of feed versus food. Thus, thirty-eight percent, or 22,630 Kt, of the total U.K. crop supply (that is, domestic production and imports minus exports) in 2010 was used for animal feed. Eighty-seven percent of all barley is destined for animal feed, while about ninety-three percent of all soya beans (the great bulk from Latin America) is also used for feed. About fifty-five percent of the total cropland footprint for U.K. feed, or about 2619 kha, was located overseas in 1987, and this increased to sixty-four percent, or 3293 kha, in 2010.

De Ruiter et al. [34] find that the total cropland footprint of U.K. food supply increased between 1987 and 2010. Thus, the cropland footprint for both feed and food in 2010 was 8833 kha compared to 8406 kha in 1987. This suggests increased carbon intensity of production since grassland-fed livestock are less carbon-intensive than grain-fed livestock. These crops have been sourced increasingly from abroad, both in respect of crops for feed and for food. Thus, in 1987, the domestic share of the cropland footprint for feed and food was about forty-five and forty-two percent, respectively, and this share decreased to thirty-six and thirty-eight percent, respectively, in 2010. Thus, sixty-four percent of feed crops and sixty-two percent of food crops were imported into the U.K. in 2010, representing increasing 'carbon leakage', especially in the case of feed since this has tended to displace the more productive sector of grass-grown domestic livestock production in the U.K. In other words, the main exporters to the U.K. of ruminant feed products have intensified their production, while the U.K. has seen a commensurate extensification (de-intensification) of grass-based ruminant production. This suggests that some environmental gains through extensification of grass-based ruminant production in the U.K. have been secured at the expense of 'carbon leakage' overseas, especially to Latin America, the primary source of U.K. soya. The U.K. has also witnessed a slight decline in red meat (ruminant) consumption, but this reduction has been matched by an increase in poultry and pork consumption, both fed almost exclusively on crop feed. While these livestock types produce less GHGs in production than ruminant equivalents, they are even more dependent on the importation of feed crops, especially soya, thus sustaining 'carbon leakage' to the global South [34].

The land required overseas to meet the U.K.'s annual demand for soya between 2016 and 2018 was on average 1.7 Mha, or an area similar to that of Wales [WWF and RSPB 2020]. Sixty-five percent of the soya land footprint was located in Argentina, Brazil, and Paraguay during this period, all countries classified by the authors as high or very high risk.[4] Fifty-six percent of the U.K.'s soya imports between 2016 and 2018 were in the form of soymeal, a prime ingredient of animal feed and increasingly associated with high protein diets. The authors' data indicate that at least seventy-five percent of all imported soya is either embedded in imported meat, eggs, and dairy or is used for animal feed [35].

Again, in 2019 [36], U.K. imports of soya amounted to over 2000 Kt, forty-two percent of which came from Argentina, twenty-seven percent from Brazil, and eight percent from Paraguay. Some forty percent of this amount (1000 Kt) came without any sourcing requirements at all concerning sustainable production in relation to deforestation/conversion certification criteria. A further twenty-three percent, however, derived from quite dubious 'book and claim' certification, whereby production, which may well be unsustainable, is supposedly 'offset' by environmental 'credits' purchased elsewhere. Only nine percent of soya production outside 'low risk' countries (such as the USA and Canada, which have historically destroyed their native vegetation) can be unequivocally attributed to sustainable production on the basis of 'physical certification'. Thus, some sixty-three percent of the seventy-seven percent of U.K. soya imports from Latin America is likely to be closely linked to climate change enhancing LUC through the destruction and degradation of biodiverse carbon sinks, in addition to the soil-degrading ($CO_2$-releasing) and fossil fuel-dependent character of the capital-intensive production process itself. To this it is necessary to add the fossil fuel consumption entailed in transportation and processing of soya.

### 3.3. Ecological Imperialism

The very considerable 'carbon leakage' of the U.K. food system (some fifty percent of GHG emissions) implies that the U.K. is engaged in a relation of 'ecological imperialism' with supplier states. This is because the latter, with the complicity of agro-exporting oligarchies in the (semi-) peripheral state–capital nexus, are subordinated to the consumption demands of this 'core' state, entailing the displacement of peasant/indigenous populations and with them the potential for sustainable agroecological food production [10]. This process of displacement generates 'ecological surplus', through 'appropriation by dispossession'[5], on the basis of the 'mining' of socio-natural 'capital' nurtured by those non-capitalist populations in the form of diverse semi-natural biomes and organically cultivated soils (principally through Surplus Extraction Mechanism 2; see below). It also entails the externalisation of costs onto the supplier country in the form of compromised food security and increased precarity for displaced populations, degraded ecosystems and soils, and the exacerbation of regionalised climate change impacts in the form of droughts, heat waves, and destructive storms [10]. All these impacts contribute to the global climate crisis through the destruction of carbon sinks and the displacement of carbon sequestering agroecosystems by fossil fuel-powered agri-business. Such externalised costs, a measure of the 'spatio-temporal fix' of ecological imperialism, should, of course, be borne by the U.K. itself or, better, not generated at all, the latter feasible only by recourse to agroecological production as part of degrowth with equity (see below).

Within the U.K. itself, agriculture, in 2018, produced only fifty-three percent (by value) of the food consumed in the country, a decline in 'self-sufficiency' in relation to the heyday of 'political productivism' and reflecting the trend towards neoliberalisation in the food system [1]. Land distribution is highly inequitable, with less than one percent of the population owning half the land in England, for example, with landownership even more concentrated in Scotland [37–39]. There were 217,000 farm holdings in the U.K. in 2017, with around twenty percent of these comprising 'very large' holdings (over hundred ha in area) and using the majority of land (seventy-six percent), while forty-eight percent of farms are 'small' (less than twenty ha), farming just four percent of land [40,41]. Production output is also very unevenly distributed—thus, in England in 2017, a small number of large

farms (only seven percent) produced fifty-five percent of output by value on only thirty percent of farmed area [40].

These larger farms are highly capitalised and usually capitalist enterprises (although commonly centred around family labour), and achieve very high productivity (the ratio of labour input to output), but only with massive quantities of climate change-inducing fossil fuel, synthetic fertiliser, and agrochemicals, together with extraordinarily expensive equipment and infrastructure, again dependent on fossil fuel. Agrochemicals derive, of course, from oil, while immense amounts of fuel and electricity are required to synthesise artificial fertilisers from natural gas. Most fertiliser is made from ammonia ($NH_3$), which is itself produced in factories, whereby nitrogen from the atmosphere is synthesised with hydrogen atoms extracted from fossil fuels at high temperature and under high pressure. Being highly soluble, synthetic fertiliser that is not taken up by plants is washed into surface or groundwaters, causing huge eutrophication problems exacerbated by the loss of soil structure and organic soil content through the persistent application of artificial fertilisers themselves (with oxidation of soil organic matter itself being a significant source of $CO_2$ emissions). The excess synthetic fertiliser that is not leached into water is converted by bacteria into nitrous oxide ($N_2O$), a GHG that, as noted, is many times more potent than $CO_2$.

Once the raw ingredients have left the farm, another long chain of energy consumption begins, comprising processing, packaging, transport, retail, cold storage, cooking in homes and restaurants, and, lastly, waste disposal [30]. Of these 'direct emissions' or supply chain emissions (primary production to RDC and RDC to consumption but excluding LUC), primary production generates fifty-six percent, with $CH_4$ and $N_2O$ accounting for more than half of these [31]. $CH_4$ is a more potent GHG than $CO_2$ (although much shorter-lived) and is produced by ruminant livestock, especially when fed on grains rather than grass, while manure from these animals also releases $CH_4$ and $N_2O$. The ecological inefficiencies and destructiveness of raising livestock fed with agro-industrially produced grains (especially soya, which embodies LUC impacts as documented above) are thus immense. Overall, sixty percent of the grain grown in the U.K. is fed to animals [41,42], while some eighty-five percent of agricultural land is devoted directly or indirectly to livestock production [32,34,43]. As much land overseas is used to support the U.K. grain-based livestock production system as is used in the U.K. itself [31,34].

*3.4. Surplus Extraction Mechanisms*

We can translate these highly unsustainable parameters of the U.K. food system into three Surplus Extraction Mechanisms, all subsumed within the category of 'imperialist rent', that is, the above average or extra profits realised as a result of inequality between the North and South in the global capitalist system [44]. The three main drivers of the U.K. economy, namely banking/services (finance capital), mining, and fossil fuel extraction, are all predicated on 'imperialist rent'. First, U.K. finance capital has invested in labour-intensive and polluting industries that have re-located to the global South during the neoliberal era especially, and here, 'imperialist rent' is founded on the super-exploitation of labour power through Surplus Extraction 1. This is founded on the huge wage differentials that exist between the global South and global North, despite comparable productivity levels of labour power between the two. This differential is key to the super-exploitation of labour power in production in the South and the differential location of high consumption in the global North, a phenomenon known as 'labour arbitrage' [45]. This surplus extraction mechanism occurs primarily through the export of industrial commodities from the South to the North. This transfer of surplus value from global South to North helps to support the levels of affluence in the latter, on which meat- and dairy-rich diets, with their adverse climate impacts, are predicated.

Second, the key mining and fossil fuel extraction sectors of the U.K. economy realise 'imperialist rent' primarily through Surplus Extraction 2. This is undertaken principally through capital-intensive extractive processes with little use of human labour power, reliant

on the 'ecological surplus' embodied in energy-dense fossil fuels and the 'socio-ecological capital' built up by non-capitalist social systems and extracted through 'appropriation by dispossession' (land grabbing). Imperialist rent is founded, inter alia, on the failure to enforce in the global South norms of environmental regulation, rehabilitation, and social compensation that would be required in the North. Super-profits thereby generated through both Surplus Extraction 1 and 2 afford the transfer of wealth to the U.K. that enables the 'imperial mode of living', the background affluence that undergirds, inter alia, the adoption of meat- and dairy-heavy diets with their ecological inefficiencies and climate change impacts. Super-exploitation of global Southern labour power and environments, or the supply of 'cheaps', implies not only more disposable wealth in the U.K. (as in the North generally) but more consumption since the cheaper the commodity, the more of it will be consumed, varying 'elasticities of demand' notwithstanding. Super-exploited labour and environments in the global South enable the continued formation of 'labour aristocracies' and specialisation in finance/services and high end manufactures in the U.K., with the country, despite increasing wealth differentials, maintaining status as the fifth or sixth richest state globally, as measured by GDP [31]. The historical and sustained full agrarian transition of the U.K. population out of agriculture (with only 426,000 people, or 1.5 percent of the U.K. labour force, remaining in agricultural production [41]) implies that agriculture is highly capitalised and resource intensive in order to supply the high consumption demands of its generally wealthy, non-agrarian, population.

The climate change impacts of such high capitalisation and fossil fuel dependency of U.K. agriculture are exacerbated by grain-fed meat and dairy-oriented diets of the U.K. population as enumerated above. The ecological inefficiencies and climate change impacts of these 'imperial' diets are, as we have seen, to a considerable extent externalised onto the global South through 'carbon leakage', such that around half of GHG emissions associated with the U.K. food system are generated overseas. The GHG emissions thus externalised onto the global South through direct production and through land-use change (LUC), principally to supply feed for livestock consumption, operate largely through Surplus Extraction 2.

Additionally, land devoted to livestock grazing in the U.K., especially on poorer soils in the uplands, could be used to a much greater degree than at present for growing timber and fuelwood. (These areas are currently dominated by sheep production, but consumption of sheep meat contributes only minimally to the food security or calorific value of food consumed in the U.K.) Not only does this compromise potentials for greater $CO_2$ absorption in the U.K. through woodland expansion, but it also displaces timber and fuelwood production overseas to areas where it may involve LUC from forest to agriculture. Thus, the land required overseas to supply the U.K.'s demand for timber has increased threefold since 2011 (from 2.8 to 8.4 Mha), an area greater than the size of Scotland [35]. Around one-fifth of the U.K.'s overseas land footprint was located between 2016 and 2018 in high risk countries (as defined earlier), including Brazil, China, and Russia. Fuelwood is used primarily for energy generation, and demand in the U.K. has increased from an average of twenty-two percent of total imports in 2011 to thirty-two percent in the period 2016–2018. This increase is likely to be linked to policies designed to increase the share of renewable sources in the U.K.'s energy matrix. Although well intended, these policies fail sufficiently to assess the carbon impacts of biofuels [35].

Finally, the U.K. food system (like the U.K. economy as a whole) emits far more GHGs than are sequestered in the U.K. We have suggested that, in total, U.K. GHG emissions are around 850 $MtCO_2e$ [30]. Net emissions in 2017 were 460 $MtCO_2e$ [46], implying that the U.K. emits some 460$MtCO_2e$ more GHGs than it sequesters. (This net emissions figure does not, however, include 'carbon leakage' due to overseas GHG emissions associated with the U.K. economy, while much of the carbon supposedly sequestered by U.K. vegetation does not in reality occur because most of this is cancelled out by the oxidation of degrading peatlands [46]. Consequently, net emissions are likely to be considerably higher in reality than the above figure suggests.) This indicates, in turn, that the U.K. is reliant

upon extra-territorial GHG sinks to mitigate the climate change impacts flowing from its GHG emissions. These it receives in large measure gratis, thus significantly lowering (externalising) the costs to the U.K. that would be borne were the country to sequester its own GHG emissions. Since such carbon sinks, in terrestrial terms, are located differentially in the global South (in the form of differentially intact ecosystems and agroecosystems sustained by indigenous and peasant production), this represents a further mechanism of surplus extraction from the South to the U.K. (and the global North more generally) since the South is bearing the cost of sequestration that should be borne by the U.K. The sad irony here, of course, is that Surplus Extraction Mechanisms 1 and 2, through destruction and degradation of these ecosystemic sinks, are actively undermining the capacity of the global South to continue to sequester GHGs, not only compromising Surplus Extraction 3 for the U.K. (and the North) but, more importantly, posing dire consequences for the future of the Earth in terms of accelerated climate change.

## 4. Biodiversity Impacts of the U.K. Food System

### 4.1. Generic Issues and Structural Causality Underlying Biodiversity Decline

It is important to recall at the outset that a large proportion of the most valued biodiversity habitats and landscapes in the U.K., as in Europe as a whole, has arisen from pre-industrial agrarian management of the, consequently, *semi-natural* environment over a very long period [47]. Thus, much of the biodiversity resource of the U.K. depends for its survival, and a fortiori for its flourishing, upon the continuation or re-adoption of low-input farming systems and practices. This comprises a co-evolutionary relationship between pre-industrial farming and the sustainable management of semi-natural habitats within a framework of *land sharing* rather than *land sparing*. The latter comprises a dichotomous framework in which *de-naturalised* agriculture is given over to productivism, while the residue of non-farmed landscape is abandoned to *de-socialised* re-wilding. The co-evolutionary relationship between farming and the semi-natural environment began seriously to erode with the rise of capitalism and industrial agricultural techniques, a trend that, as we have detailed, accelerated out of all recognition following the Second World War. The post-war period has witnessed steep declines in the area of semi-natural habitat and in the number and range of characteristic farmland bird, invertebrate, and plant species. Survival of these habitats and species now occurs (increasingly scarcely) *despite* rather than (as before) *due to* agricultural practices [20]. (We focus in this section on biodiversity impacts of productivist agriculture in the U.K., while acknowledging, as detailed in the previous section, the significant adverse impacts of the U.K. food system on biodiversity overseas, especially in the global South.)

As with climate change impacts, this massive acceleration in the rate of biodiversity loss and decline may be attributed *structurally* to the impacts of capitalism as a general tendency and, more specifically, to a particular model of capitalism that we have identified as 'political productivism' (and its tendential neoliberal successor 'market productivism') [20]. The environmental impacts of productivism can be enumerated as a series of generic issues:

- Loss and fragmentation of semi-natural 'infield', traditionally grazed habitats through agricultural 'improvement' (application of synthetic fertilisers and herbicides) or conversion of these to arable land;
- Overgrazing of semi-natural habitats, primarily in the uplands;
- Loss or mismanagement of 'interstitial' habitats, for example, hedgerows, field margins, ditches, ponds, etc.;
- Drainage or drying out of wetland habitats due to water over-abstraction;
- Pollution and eutrophication of surface and groundwaters leading to loss or degradation of aquatic ecosystems;
- Loss of crop rotations and arable–pasture mosaics leading to severe reduction in characteristic farmland species;
- Shift from spring-sown to autumn-sown cereals leading to loss of nesting sites for characteristic farmland bird species;[6]

- Generalised application of pesticides leading to loss of arable weed species, invertebrates, and thereby food sources for other wildlife groups;
- Generalised application of synthetic fertiliser leading to the loss of degradation of semi-natural vegetation, decline in the organic content and structure of soils, and eutrophication of ground and surface waters [20,47–52].

These generic issues can be linked causally to the essential features of the capitalist and productivist impulses embodied in 'appropriationism' and 'substitutionism', a relationship that we can nominate as *structural causality* [4].

As a result of these generic or structural impacts of capitalist productivism, semi-natural 'infield' habitats have been pushed to the margins of agrochemically based agriculture, subsisting as a residual resource peripheral to most farming systems [20]. Only in the uplands, where physical constraints have prohibited the widespread application of synthetic fertilisers and pesticides, do semi-natural habitats still comprise integral elements of farming systems [48]. In much of the lowlands, however, semi-natural 'infield' habitats survive typically as mere fragments within an otherwise ecologically impoverished farming landscape. Even 'common' species characteristic of more productive farmland (that is, traditionally farmed 'artificial' infield habitats such as arable and grass leys) have exhibited alarming declines over the course of the productivist era [49]. Freshwater habitats continue to suffer decline and loss through nutrient pollution and water abstraction from agrochemically based farming practices. Rivers in catchments where large-scale chicken/egg production is located (birds fed principally on imported soya) suffer increasing eutrophication and ecological degradation through the spreading of manure on land in quantities that the soil cannot absorb. Surplus nutrients, principally phosphates, are then washed into adjacent watercourses. In the uplands, habitat deterioration rather than outright loss has been the norm, the result most frequently of ecological overgrazing by livestock, principally sheep [50].

The current agricultural and environmental policy framework in the U.K. (these frameworks differ slightly, although are diverging increasingly, between the constituent countries' administrations to whom agricultural and environmental policy is devolved) affords a proportion of this residual resource a modicum of legal protection and conservation management by means of statutory regulation and/or environmental land management schemes (ELMS). As we shall see below, Brexit U.K. is introducing a new suite of ELMS following the discontinuation of CAP-derived schemes—these will be configured in a more purely neoliberal way than hitherto in conformity with the neoclassical economic theory of 'market failure' and the payment of so-called 'public money for public goods'. We will summarise the deficiencies of this theory and approach below. For the moment, it is sufficient to note that while environmental regulation, cross-compliance prescriptions attached to continuing direct farm payments (Basic Payment Scheme), and existing ELMS have slowed the decline of biodiversity in the farmed landscape (in other words, decline would have been worse without these sources of mitigation), alarming declines in the extent, quality, and numbers of both habitats and species continue to characterise the U.K. agricultural environment. In 2019, the *State of Nature* report indicated the following:

*A wide range of changes in agricultural management in recent decades has led to greater food production but these changes have also had a dramatic impact on farmland biodiversity. For example, populations of farmland birds have more than halved on average since 1970, and similar declines have been seen in many other taxonomic groups. Targeted wildlife-friendly farming, supported by government funded agri-environment schemes, can help halt and reverse these declines, but to date the only successes have been for rare and localised species. The area of land receiving effective agri-environment measures may have helped slow the decline in nature, but it has been insufficient to halt and reverse this trend.* [51]

The most recent *State of Nature* report published in 2023 [52] noted the unfortunate continuation of this general downward trend in the abundance and distribution of biodiversity, especially of that component not adapted to productivist environments:

> *…the UK's nature and environment continues, overall, to decline and degrade…the size of response and investment remains far from what is needed given the scale and pace of the crisis.* (p. 3)

> *The best available information suggests that nature-friendly farming needs to be implemented at a much wider scale to halt and reverse the decline of farmland nature.* (p. 7)

> *If we are to halt and reverse biodiversity decline we need not only to increase our efforts towards conservation and restoration, but also to tackle the drivers of biodiversity loss, especially in relation to our food system. That means making our food more sustainable and nature-friendly and adjusting our consumption to reduce demand for products that drive loss of nature.* (p. 9)

These declines, again, can be attributed to structural causes arising from capitalist productivism. In response to these drastic declines in biodiversity, there is an urgent need to firstly conserve and enhance the remaining resource of semi-natural habitats through site buffering, linkage, and re-creation and to secondly address the decline in 'common' habitats and species in the 'wider countryside'. We will argue that this can be secured only through an integrated, *land sharing* perspective premised on the cessation of capitalist productivism (privileging the production of exchange value through farm capitalisation and intensification) and the adoption of the agroecological production of use values through 'nature-based' solutions [7,53].

### 4.2. Neoliberalism and the Land-Sparing Approach to Agri-Environmental Policy

The currently prevailing model of biodiversity conservation in the U.K., one that will continue and be reinforced with the adoption of neoliberally configured ELMS, is one in which nature is 'sequestered' on special sites/areas and accorded a role subordinate and opposed to the capitalist productivist impulse to maximise exchange value (growth) through capitalisation and intensification. In this de facto land sparing approach, biodiversity conservation is undertaken on a site-by-site, species-by-species basis and awarded a separate (usually paltry) budget for a series of discrete conservation activities that are juxtaposed to and must match the opportunity cost of the productivist capitalist enterprise [20]. Indeed, payment rates for ELMS are calculated not on the basis of the intrinsic value of the conservation resource but rather on the basis of 'profit foregone', that is, the amount of money that could be made by the capitalist enterprise were the biodiversity resource in question to be destroyed through 'improvement' (in other words, the 'opportunity cost' of foregoing productivist farming practice). The two aspects of this policy framework, a spatial/sectoral dichotomy between environmental and capitalist farming concerns and the expectation that biodiversity will be conserved only if the opportunity cost of agricultural 'improvement' is met, reflect both the productivist foundation of the state–capital nexus and, within this, the absolute property rights that farmers have been awarded to enable them to claim 'compensation' for not destroying nature [54].

The bankruptcy of this approach has been exposed over the last few decades by the continuing and inexorable decline of the U.K.'s biodiversity resource, exacerbated by persistent indirect subsidies to productivism in the form of the Basic Payment Scheme (only weakly linked to environmental outcomes) on the one hand and austerity-hit budgets of environmental agencies on the other. This continuing and alarming decline exposes not only the inadequacy of the land sparing approach to biodiversity conservation (nature cannot be conserved effectively on an isolated or fragmented basis or in relation to individual species alone—see [53,55]) but also the impossibility of effective biodiversity conservation when the structural causes of decline arising from capitalist productivism, whether 'political' or 'market' productivist in orientation, remain in place. Rather, change is required towards environmental (and social) sustainability in the character of that economic activity

itself [20]. This might be defined as 'strong sustainability' [56]. This means that sustainability will not be secured through mitigating (in effect 'buying off') unsustainable agricultural practices, an approach embodied in the prevailing model of voluntary environmental incentive schemes (and to be perpetuated in the new ELMS), but will need to be secured by addressing (resolving) the structural causes of generic impacts, whether these derive from 'political' or 'market' productivism. At base, this means rendering food production itself agroecological so that there is no longer a dichotomy between food production and ecological sustainability.

A structural or generic issues analysis, a land sharing or whole countryside approach, and strong sustainability (a sustainable social metabolism) are mutually defining since each derives from or implies the other—a holistic and dialectical relationship between theoretical cause and political praxis. A land sharing and whole countryside approach has as its objective not only the conservation and enhancement of semi-natural habitats subsisting at the margins of productivist farming but, additionally, the transformation of its 'infield' practices into those of an agroecological orientation. The latter objective seeks to conserve and enhance not only characteristic biodiversity but also the resources of soil, water, and atmosphere and to provide nutritious food equitably to all citizens [53,55]. In short, a land sharing approach involves farming that satisfies the requirement for the *joint and sustainable production* of food, biodiversity, and soil/water/atmosphere. This simultaneous and equal concern for all the dimensions of sustainability is embodied in the principles of political agroecology and food sovereignty [4,7]. Sadly, agricultural and environmental policy, especially in England, the home of doctrinaire neoliberalism, is moving in precisely the opposite direction. The Agriculture Act of 2020 embodies the proposed abandonment of any pretension towards the joint management and multifunctional delivery of both food *and* nature; rather, the former is to be left to the tender mercies of the international 'law of comparative advantage', while state subvention and management is to be confined to the latter, shoring up the supposed 'market failure' of environmental provision through 'public goods for public services' [1][7].

## 5. Dietary and Food Security Impacts of the U.K. Food System

### 5.1. Dietary Impacts

If the climate change and adverse biodiversity impacts of the U.K. food system are highly concerning, so too are the socio-economic inequalities in diet and food access generated through its operation. Thus, the annual National Diet and Nutrition Survey produced by Public Health England and the Food Standards Agency demonstrates that adults on low incomes are more likely to have diets high in sugar and low in fibre, fruits, vegetables, and oily fish. Children from the least well-off twenty percent of families consume approximately thirty percent fewer fruits and vegetables, seventy-five percent less oily fish, and seventeen percent less fibre per day than children from the most affluent twenty percent [57]. These differences in diet have an important influence on health inequalities that correlate with socio-economic status. Populations resident in the most deprived decile of neighbourhoods are almost twice as likely to die from preventable causes by comparison to those in the wealthiest decile: they are 2.1 times more likely to die from preventable heart disease; 1.7 times more likely to die from preventable cancer; and three times more likely to have tooth decay at the age of five; children are nearly twice as likely to be overweight or obese at the age of eleven [58]. Since 2010, after which economic austerity policies began to be introduced, life expectancy has gone into reverse in the most deprived areas. Thus, women in the most deprived ten percent of neighbourhoods in England now die 3.6 months younger than they did in 2010, and their life expectancy is 7.7 years less than that of women in the wealthiest ten percent. The differential for men is 9.5 years [58,59]. For healthy life expectancy, there is an even greater disparity of nineteen years between the ten percent of poorest and richest [58,59].

The modern, industrial food diet of cheap 'junk' food (highly or ultra-processed food) possesses the singular but perverse quality of generating obesity and poor nutrition

simultaneously. Thus, children in the poorest areas of England are both more overweight and significantly shorter at age ten–eleven than their peers in the richest areas [60]. The average five-year old in the U.K. is now shorter than his/her peers in nearly all other high-income western countries [61], an indication of the adverse health impacts of neoliberal austerity policies in the U.K. by comparison to more interventionist policies of most states elsewhere in the global North. Obesity can and does co-exist with outright hunger; the same households that eat poorly may find themselves unable to eat at all, a phenomenon that has been increasing with the recent 'cost-of-living' crisis in the U.K. caused by the Ukrainian conflict and commensurate increases in the cost of basic food items through inflationary pressure on artificial fertilisers (exacerbated by Brexit and exposing the vulnerability of the 'globalised' U.K. food system to external disruption [1]). Thus, data collected by the Department for Work and Pensions in 2019 found that, even before the COVID-19 pandemic, four percent of U.K. families experienced disrupted eating patterns or were obliged to cut back on food due to poverty [62]. Among recipients of Universal Credit (a general welfare payment), this proportion rose to twenty-six percent [62], a percentage that has increased still further with COVID-19 and the subsequent 'cost-of-living' crisis.

Sadly, poverty reinforces unhealthy food consumption, if and when such food is affordable and available. Unhealthy food is cheaper per calorie than healthy food and is more readily available in poorer neighbourhoods. There is a clear correlation between poverty and the density of fast-food outlets, for example [63]. Over three million people in the U.K. cannot reach any food stores that sell raw ingredients within fifteen minutes by public transport (which in the U.K. continues to deteriorate, with cancelled bus routes and less frequent services), and forty percent of lowest income households lack access to a car [63]. The lack of easy access to fresh ingredients or, in the case of nearly a million people, to a fridge in which to keep perishables, together with the increase in the cost of electricity and gas, compounded by the time and effort required to decide upon a menu and prepare a meal, act in tandem as strong deterrents to those with fewest resources and skills to cook 'from scratch' [31]. More serious still, access to affordable food of any kind is now beyond the means of a growing minority of the population—such 'absolute' food poverty is an increasing reality for those on low incomes or social security in the context of the 'perfect storm' of precarious incomes, a reducing welfare state, and inflationary pressure on food prices. As a result, the U.K. now has some two thousand food banks run by charities to supply free food to people in need. Nearly fifty percent of families with three or more children are now living below the poverty line, while the most recent cut in Universal Credit introduced in 2021 was predicted to drive a further half a million people into poverty and push the child poverty rate to one-third of all children [62,64].

The Food Foundation monitors moderate or severe hunger and malnutrition in the U.K. (food insecurity defined as insecure access to adequate amounts of nutritious food). Moderate or severe food insecurity are defined, more specifically, as the number of people in the previous month who had smaller meals or skipped meals; had been hungry but not eaten; or had not eaten for a whole day—each because of lack of access to or inability to afford food. In June 2023, the Food Foundation found that nine million adults in the U.K., or seventeen percent of households, experienced moderate or severe food insecurity, a massive rise from 7.3 percent in June 2021 [65]. Nearly one-quarter of households with children experienced food insecurity [65]. As suggested above, moderate or severe food insecurity in the U.K. is associated with obesity, since people who cannot afford or lack access to healthy and nutritious food eat unhealthily. Far from seeking to help low-income families to escape these life-debilitating disadvantages, however, the neoliberal policies of the incumbent U.K. government are simply exacerbating the conditions underlying dietary inequality and food poverty. In short, the wider social inequalities of neoliberal Britain are reflected in and exacerbated by inequalities of diet and food poverty.

*5.2. Food Security Impacts*

Turning to the issue of self-sufficiency and food security, we may define self-sufficiency as the ability to feed a nation from its own produce rather than from imports. Food security, for its part, has been defined as the ability to feed a nation *at a reasonable cost*, irrespective of the source of food or the manner of food production, even in the face of future shocks such as massive harvest failure or a general crisis of agricultural production caused by, for example, climate change [31]. Sadly, however, this definition, one operationalised by historical and contemporary U.K. governments alike, directs virtually no attention to the ecological costs entailed in supplying food to the nation 'at reasonable cost'—in other words, 'at reasonable cost' usually means 'at huge cost to the environment', 'cheapness' being predicated on ecological (and social) cost externalization, both domestically and overseas [1]. Stated otherwise, it is the 'hidden' subsidy afforded by ecological (and social) cost externalisation that enables food commodities, under the principle of capitalist 'comparative advantage', to be supplied 'at a reasonable cost' to consumers [4]. 'Food security', as currently configured, therefore generates longer-term food insecurity by undermining the ecological and biophysical basis of agricultural production [1]. Moreover, a capitalist food system cannot assure general access to food since this depends on the ability to pay—capitalist food producers have to balance 'cheapness' (affordability) with the imperative to generate profit. The state–capital nexus attempts to mitigate and balance these contradictions—the need to ensure the availability of food for consumers 'at reasonable cost' and the desire of capitalist producers to realise a profit—but, as a capitalist state, is ultimately constrained by capital's demand to realise surplus value as the 'bottom line' [17].

Food sovereignty and political agroecology propose, by contrast, to extricate food production, distribution, and consumption from capital's grip and place them in the hands of democratically controlled governance mechanisms to secure ecological sustainability and equality of access to wholesome food and to the means of food production [4,7]. National food self-sufficiency does not guarantee food security; but where food sovereignty and political agroecology are introduced within any polity, self-sufficiency is the only means of ensuring, so far as is feasible in relation to 'indigenous products', that food is produced, distributed, and consumed in conformity with the principles of ecological sustainability and social equity. While it is important to ensure that alternative supply is available in the event of harvest failure or other local crises, it is vital to repeat that 'food security', as currently defined and enacted by the U.K. government, is achieved only at cost of food insecurity over the longer-term as the ecological foundations of production are eroded and climate change accelerates [1].

Indeed, the U.K. state–capital nexus has, since the emergence of industrial capitalism in the 1840s, sought to deliver 'food security' on the basis of international capitalist supply, premised on the principle of 'comparative advantage' or 'least cost'—a principle that asks few questions concerning the ecological or social sustainability of food supply (only the two world wars and the aftermath of the latter have proven exceptions to the doctrinal dominance of comparative advantage) [4]. Consequently, food self-sufficiency in the U.K. fell progressively following the abolition of the protectionist Corn Laws in 1846 [1,4,18], reaching a nadir of thirty percent self-sufficiency on the eve of the Second World War. Wartime blockades and concerted efforts to increase domestic production saw self-sufficiency rise to seventy-five percent by the end of the Second World War (1945) [1]. The post-war period saw the introduction of 'political productivist' policies to boost self-sufficiency as discussed earlier in this paper. Thus, by the mid-1980s, when CAP subsidies and tariffs were at their zenith, U.K. self-sufficiency reached a peak of around eighty percent [66]. Subsequently, the fiscal costs and environmental impacts of 'political productivism' led the EU to reduce commodity support and tariffs; U.K. governments since Margaret Thatcher have been keen advocates again of freer trade, 'comparative advantage', and, consequently, 'globalisation' in agriculture, as with most sectors. U.K. domestic self-sufficiency has declined since then, a decline that has accelerated with Britain's departure from the EU and, on the basis of current trends and policies, is set to continue on this trajectory. Indeed, the present U.K.

government has such confidence in the operation of the capitalist market and the principle of 'comparative advantage' that it no longer sets a target for the amount of food that the U.K. should grow to feed itself [31]. It relies, rather, on two methods to assess national 'food security'.

First, it conducts occasional reviews to assess whether the U.K. possesses the means to restore necessary self-sufficiency in the event of food supplies from other countries being cut off completely. In 2009, DEFRA conducted a U.K. Food Security Assessment [67], concluding that the U.K. already grows much more of our own food than was the case before the Second World War—the U.K. would therefore be better placed than seventy years previously to restore self-sufficiency if needed (although self-sufficiency has continued to decline since that assessment was conducted). The 2009 assessment also indicated that the shortfall in self-sufficiency could be made good by a shift from livestock production to grains and vegetables. Presciently, from the perspective of this paper and the climate emergency-inspired need to transition to agroecology, the assessment suggested that the maximisation of calorie production would require a dramatic reduction in livestock production, with all crop production being used where possible for human consumption rather than for animal feed [67]. DEFRA estimated that, in such a scenario, the U.K. could produce more than enough calories per person per day (although this would be undertaken using synthetic fertilisers and fossil fuels of course—see below for discussion of agroecological transition and abjuring reliance on fossil fuels).

Second, DEFRA also conducts internal monitoring of food security, these reports being shorter but more wide-ranging than the 2009 Food Security Assessment. They assess the risk of various disruptions across the food system: how global harvests might change due to global heating and other stressors, the geographical diversity of U.K. food imports and the degree of exposure of the U.K. to harvest failure in any one region of the globe, etc. The 2020 Agriculture Act formalised DEFRA's food security reviews, creating a statutory duty to publish such a report at least every three years [31]. As noted earlier, however, the Act also cemented a commitment to increased reliance on cost-externalising 'comparative advantage', thus structurally reinforcing both the U.K.'s commitment and vulnerability to market-defined 'food security'—in effect, food insecurity [1].

In this way, industrial 'market' productivist agriculture, exacerbated by 'globalisation', continues to cause climate change, together with the other severe disbenefits enumerated above, in turn threatening longer-term food supply. Yet, rather than seeking to address this fundamental cause at home by increasing sustainable food self-sufficiency, the present U.K. government, in its quest for 'cheap' food through 'comparative advantage', is bound upon a course of increasing unsustainable 'globalisation' by its commitment to signing FTAs with countries with lower environmental standards than the U.K.'s. As the National Food Strategy [31] points out, it is fairly pointless trying to build a low-carbon, nature-friendly, and socially equitable food system in the U.K. if it is then undercut by imported food produced to lower standards. And yet, this is precisely what the present U.K. government is in the process of doing.

In the next section, we examine in more detail the policy discourses of the major politico-economic constituencies that are attempting, variously, to obfuscate, mitigate (symptom manage), or resolve the contradictions of the U.K. food system delineated above. This examination will serve to paint a picture of the current politico-economic context dominated by various shades of capitalism, which any transition to agroecology and degrowth in the U.K. must unavoidably confront.

## 6. Contested Policy Discourses Surrounding Agrarian Climate Change, Biodiversity Loss Mitigation, and Food Security Strategies

*6.1. Net Zero: 'Societal Project' or 'Flanking Measure'?*

Following mounting international pressure from the IPCC, and specific recommendations from the advisory U.K. Climate Change Committee (CCC) to address comprehensively the incontrovertible causes and impacts of anthropogenic climate change wrought by GHG

emissions, the U.K. government under a Conservative Party administration announced a target of net zero for such emissions by 2050. The change to legislation came into force on 27 June 2019 and imposed a legally binding target on U.K. governments to achieve such a target by the stated date, thereby amending the Climate Change Act of 2008 (which had set a target of eighty percent reduction in GHG emissions compared to 1990 levels). Net zero means that the U.K.'s total GHG emissions will, by 2050, need to be equal to, or less than, the emissions the U.K. removes from the environment. It is proposed that this can be achieved by a combination of emission reduction and emission removal. GHG emissions can be removed by the natural environment (natural sinks/sequestration) or by using, the U.K. government asserts, technologies like carbon capture (usage) and storage (CC(U)S) [46], even though these are as yet unproven technologies in terms both of feasibility and safety.

Given that this legislation has been passed by a Conservative administration whose neoliberal or 'hegemonic' policies hold out little actual hope of meeting this emission reduction target (as is intimated above and further substantiated below), it is reasonable to surmise that the real purpose of the commitment is to construct a legitimating 'flanking measure' [15] to neutralise potential oppositional discourses, especially from influential environmental civil society groups, and to persuade doubters that something meaningful is being undertaken to avert climate crisis [68]. The adoption of a legally binding and very concrete target certainly sends a clear legitimating message of belief that current and proposed market dependent policies and techno-optimistic imaginaries will deliver on the emissions reduction objective if only 'rational' price signals can be liberated from the dead-hand of the state—and here 'liberation' from the 'sub-hegemonic' and 'state-interventionist' instincts of the EU is construed as a clear opportunity for neoliberals to demonstrate the putative efficiency of free markets in securing the 'optimal allocation of resources' in relation, inter alia, to climate change mitigation and biodiversity conservation.

This combination of immovable target and techno-market optimism appears designed, then, to deflect competing discourses entailing greater levels of market constraint and state interventionism such as are characteristic of the 'sub-hegemonic' narrative, for example. It is certainly true that there are many amongst the Conservative Party, the civil service, and neoclassically trained economists in environmental organisations who genuinely believe in the beneficence of the market (see, for example, [69]) and in its capacity, with the proviso of 'public money for public goods' and certain regulatory safeguards, to secure the required 'net-zero' emissions and improvement in the status of biodiversity. Currently, then, a combination of neoliberal and 'quasi-neoliberal' (that is market-optimist and market-oriented climate emergency discourses as defined earlier and discussed below) may be said to have entrenched hegemonic status in the U.K. Parliament, in state departments, in agri-food capital, amongst larger farmers, and amongst mainstream environmental pressure groups. To the extent that this 'combined' neoliberal discourse coheres and appears unassailable, it may indeed be considered to act as a 'societal project' or 'semantic fix' [68,70]. However, as post-Brexit optimism wains, and dissension again raises its disruptive head, it is perhaps more helpful to speak of the net-zero target as a 'flanking measure' [15], since this term suggests continuing underlying tension beneath this provisional hegemony. Booth [57] suggests the following:

> Attributing the "post-political" [that is, uncontested hegemony] *label to a field wherein even relatively aligned institutions are in tension is overly simplistic. The unfolding of these tensions in the coming decades must be attended to in a way that does not reify a monolithic state-capital nexus, but instead acknowledges the dynamism and lived nature of institutions and the class interests they represent, modulate, and materialise. These fissures are currently predominantly discursive in nature [due to provisional hegemony], but as the British countryside is reshaped in various forms by impending socio-ecological change, they will become socio-material gulfs.*

Booth [68] also notes that the U.K.'s net-zero by 2050 ambition has stimulated food system actors, additional to those pursuing a neoliberal course, to propose various 'imagined

pathways to net-zero agriculture', pathways that may be variously seen as complementary to neoliberal discourse ('post-productivism' and 'alter-hegemony'), in tension with it ('sub-hegemony' or 'neo-mercantilism'), or diametrically opposed to it ('counter-hegemony' or 'radical' food sovereignty). We now examine the discursive content of these pathways, the class or interest group complexion of their proponents, and their policy proposals for the agri-food sector to secure net-zero by 2050 as well as ancillary objectives for biodiversity conservation and implications for food security.

### 6.2. Hegemonic and Quasi-Hegemonic Discourse

The first and '**hegemonic**' discourse is that of neoliberalism, conforming to the 'corporate driven, technological (optimist)' narrative identified by Borras et al. [71]. This is represented most fulsomely by many members of the Conservative Party, by state departments such as the Treasury and DEFRA, by many upstream and downstream fractions of agri-food capital, and by larger farms confident of their continued 'comparative advantage' without state subvention. However, we here combine this discourse with a 'climate emergency' narrative [71] characteristic of the U.K. CCC and certain statutory and non-statutory environmental bodies, for example. We do so because, although these bodies adopt rigorous natural scientific criteria to define the necessary parameters as the basis for policy action to mitigate climate change, together with biodiversity and natural resource decline, the recommended policy actions themselves are largely defined by neoclassical and neoliberal economic theory. Thus, while Booth [68] suggests that the U.K. CCC deploys a 'sub-hegemonic' discourse to differentiate it from the more doctrinaire neoliberalism of DEFRA, for example, we prefer to nominate the CCC as embodying a '**quasi-hegemonic**' narrative.

DEFRA may be taken as an exemplar of 'hegemonic' neoliberalism [11,12,68]. The overall pathway to net-zero that emerges from its departmental literature and statements is one reliant on 'faith in markets to solve complex socio-ecological problems' [68]. DEFRA accords great responsibility to individual farmers to attain net-zero through the prospect of innovation and the 'rationality' of the market. This basic competitive dynamo of supposedly beneficial change is to be supplemented by a measure of grant-based investment in both on-farm capacity and innovation as a means to bolster productivity improvements (the ratio of labour input to output) still further [72–74]. This stance is reiterated in an update of the Agricultural Transition Plan (January 2024) [75]. However, despite this optimism in the market, productivity improvements, and technological innovation on the part of individual farming entrepreneurs, DEFRA in actuality places central reliance on the new *publicly funded* Environmental Land Management Schemes (ELMS) to secure its net-zero vision, together with and through the mitigation of biodiversity and natural resource decline. While ELMS are legitimated according to the neoclassical doctrine of 'public goods' payments in the event of 'market failure' [13], and while disbursements are intended to be strictly delimited and competitive (with the possible exception of the Sustainable Farming Incentive (see below)), it is nonetheless the case that, in the absence of such public subvention and market intervention, there is virtually no prospect of achieving net-zero in the agri-food sector (see below).

The CCC, an exemplar of what we term 'quasi-hegemonic' discourse, is an independent, non-departmental body tasked with producing publicly available and independent expert advice on the government's efforts to meet its climate change targets [68]. Like reports from statutory environmental agencies such as Natural England, its published outputs in this area, notably *Land Use: Policies for a Net Zero UK* [41], are more technical and intended as objective analyses of potential pathways forward to ensure the government's environmental targets are secured. Further, the CCC's strategy of publishing various emissions reduction scenarios broadens its capacity to make proposals for more drastic change [41,68]. Given its formal independence, the CCC is able to recommend policy action further removed from norms of political conformity than a state department such as DEFRA. Consequently, CCC is able to deploy more of a 'climate emergency' discourse than would be feasible for DEFRA, for example, and this is indeed reflected in its unequivocal

statements indicating the need for urgent and serious change in land use to meet net-zero: 'The UK's net-zero target will not be met without changes in how we use our land. . .Current policy measures will not deliver the required ambition. . .Throughout the UK there is an urgent need for a new approach: the legislative opportunities for real change are available and should progress immediately.' [41].

Despite this invocation of expeditious and meaningful policy action to curb climate change, however, 'the CCC. . .operates within the conjunctural orthodoxy and does little to challenge existing political "common sense" around the efficacy, rationality, and desirability of markets. . .' [68]. Like DEFRA, the CCC places huge faith in the market and technological innovation to render production and productivity improvements that will enable food to be produced with fewer GHG emissions and land to be released for designated carbon sequestration projects (afforestation, peatland creation, etc.) [41]. Consequently, we consider the CCC to remain in thrall to neoclassical and neoliberal doctrine despite its acknowledgement of the biophysical causes and impacts of climate change—hence our nomination of this discourse as 'quasi-hegemonic'. This thraldom in relation to neoclassical theory is exemplified by the following definitive statement of market optimism: 'Many of the environmental goods and services that land provides do not have a private market; their positive impacts are not priced and are under-supplied by the market. This has led to historic and ongoing degradation of land, soils, and water courses and loss of biodiversity.' [41]. Thus, according to neoclassical theory and the CCC, it is not capitalism and the capitalist market that generate environmental harm as a necessary part of their operation; it is in fact their very absence. The solution is not the abolition or restraining of capitalism, therefore, but rather the further privatisation and commodification of land and nature where these are 'excludable'—where exclusion (enclosure) is not feasible, as is supposedly the case with 'public goods' provision, then 'market failure' ensues. Such 'market failure' is then to be addressed through the provision of public subvention for 'public goods', in this case the new ELMS schemes. Like DEFRA, then, the CCC places central reliance on these schemes to deliver net-zero by 2050. Below, we examine the likely efficacy of ELMS in achieving this target and other biodiversity and natural resource benefits that are claimed by the Conservative government to flow from them.

These two discourses, 'hegemonic' and 'quasi-hegemonic' neoliberalism, comprise socio-technical imaginaries of net-zero futures through central reliance on an 'open future' of market and technological optimism born of a genuine ('quasi-hegemonic') or a rather more Machiavellian and class-centric ('hegemonic') belief in the 'truth' of neoclassical economic theory. Booth [68] suggests 'that this contemporary mode of discourse-oriented target governance does not in reality attempt to "make futures". . . but only facilitate[s] the construction of future markets and market futures'. Again, there are certainly those, especially professionally trained neoclassical economists in the civil service, who appear genuinely to believe in the 'Promethean' qualities of capitalist rationality in the face of all challenges, climate change included ('quasi-hegemonic' discourse). It is nonetheless clear that the predominant and more 'Machiavellian', 'neoliberal' discourse of partisan class agents in parliament and agri-food capital is concerned primarily not to deliver the social change necessary drastically to reduce GHG emissions but rather, and as a 'flanking measure' [15], to create the illusion of government effort whilst simultaneously facilitating the 'permanence' [76] of current and increasingly 'disembedded' capitalist social–property relationships [12,68].

This lack of substantive effort to move meaningfully towards the net-zero by 2050 target is borne out by the woeful inadequacy of U.K. Nationally Determined Contributions (NDCs), which are intended to identify concrete rather than aspirational achievements and targets for GHG reduction by 2030. These NDCs currently fall far short, in terms of substantive policy actions, of the trajectory of GHG reductions that would be required to meet the 2050 aspiration, however. The NDC document (2020) identifies an ambitious target of sixty-eight percent reduction in GHG emissions by 2030 compared to 1990 levels, while a further target of seventy-eight percent reduction by 2035 compared to 1990 levels

was announced in 2021 following recommendation from the CCC [77]. The NDC, however, provides virtually no substantive policy detail as to how this target might be secured. The section of the document entitled *Food Security and Policy* [78], for example, contains no detail other than to refer to the U.K.'s Agriculture Act 2020—this Act, however, simply embodies the empty techno-market optimism and heavy reliance on ELMS delineated above. Produced in conjunction with the NDC is the Adaptation Communication [79], designed to indicate how the U.K. will adapt and build resilience to current and future climate change. This document affords a little more policy detail than the NDC but does so simply by reiterating the heavy reliance on ELMS in agricultural policy to secure GHG emission reductions in the sector: 'A cornerstone feature of future agricultural policy, the new Environmental Land Management scheme, will provide a powerful vehicle for achieving the goals of the 25 Year Environment Plan and the commitment to net-zero emissions by 2050, while supporting our rural economy. Climate change adaptation and mitigation are core aims of the Environmental Land Management scheme.' [79].

Given this centrality of ELMS in the current U.K. government's neoliberal ('hegemonic' and 'quasi-hegemonic') discursive aspiration to secure in the agriculture sector the legally-binding reduction targets it has set, including for the mitigation of biodiversity and natural resource decline, we now turn to examine the likelihood of this eventuality. The U.K. Conservative government wishes to replace, through phased withdrawal, the inherited ('sub-hegemonic') support structures of the CAP with a neoliberal system that affords no direct support to farmers since 'production and trade-related' subvention is considered anathema to the 'free play' of market forces beloved of neoclassical theory [9,12]. Public subvention, as indicated, is thus to be confined to so-called 'public goods' payments since these are the supposed result of 'market failure' and therefore receive the imprimatur of neoclassical orthodoxy. In this way, it is proposed to restrict public subvention to ELMS, of which there will be three main components. Originally, these were proposed to comprise The Sustainable Farming Initiative (to support 'environmentally sustainable farming' across the landscape), Local Nature and Recovery (to support local environmental priorities and recovery), and Landscape Recovery (to support local environmental priorities and recovery, including 'rewilding') [80,81]. The first two schemes have now transmuted into the Sustainable Farming Incentive and Countryside Stewardship, while the third retains its original name [75]. All are supposed to contribute to meeting the net-zero target through actions and practices intended to reduce and sequester GHG emissions through and together with measures to mitigate biodiversity and natural resource decline.

These are to be 'supported' by a set of regulatory standards, the configuration of which remains as yet unclear [9]. ELMS are to be configured in a way that conforms to what Tilzey [12] refers to as 'radical neoliberal' discourse, compatible with WTO 'green box' stipulations and as decoupled from agricultural production decisions as is feasible. As such, subvention will be voluntary and discretionary (competitive), meaning that, in contrast to current Pillar 1 direct payments, no 'automatic' entitlement to funding will be implied [81]. While the Sustainable Farming Incentive does appear to have features not wholly dissimilar to direct payments derived from CAP Pillar 1 (being a '"universal scheme", available to all farmers' [82]), thus representing something of a rowing back of radical neoliberalism under pressure from the likes of the NFU, the latter has nonetheless criticised the scheme as providing inadequate financial support and funding environmental actions at the expense of food production. In a response to the government's January 2024 changes to ELMS, the NFU indicated: 'With a minimum of 50% reduction in direct payments due in 2024, the tapering of payments to 2027 continues to be very concerning... It is imperative that the Sustainable Farming Incentive has sustainable food production at its core, with enough options that sit around productive farming. For this to happen it is absolutely vital that there's a better balance between policies that focus on enhancing food production as well as the environment' [83]. This statement clearly expresses the different emphases of 'hegemonic' neoliberal discourse—supporting the 'environment' through 'public good' payments whilst leaving food production (and food security/sustainability) to

be determined by 'market forces'—and 'sub-hegemonic' market interventionist discourse—supporting the 'sustainable intensification' of productivist farming to secure national food security whilst subordinating environmental actions to this overriding objective. As such, both discourses a predicated on a 'land sparing' rather than a 'land sharing' approach.

Absent the financial 'cushion' afforded by current direct payments, and given the prevalently selective and competitive character of ELMS together with their focus on 'environment' rather than supporting 'food production', British farmers will find themselves competing against adverse pressures flowing from 'market productivism' embodied in the free trade agreements (FTAs) that the U.K. government is now committed to concluding with countries that often have significantly lower environmental and social standards than the U.K., and that will, therefore, exert greater downward pressure on prices, forcing farmers to further externalise ecological and social costs [9,20]. The FTA with Australia that came into force in 2023 is symptomatic of this trend, where adverse LUC (land clearance, soil degradation, and erosion) associated with 'cheap' sheep and cattle production represents considerable 'carbon leakage' [14,84]. Even more alarming are prospective FTAs with states such as Brazil, from which the U.K. already imports five percent of its beef, with huge implications for adverse LUC and, consequently, 'carbon leakage' [34].

Under the U.K. government's post-Brexit 'global Britain' scenario, therefore, enhanced competitive pressures will oblige farmers to accelerate 'market productivism' in an attempt to compensate for increased downward pressure on prices generated by cost externalising imports predicated on 'carbon leakage'. Indeed, the U.K. is already considering 'waivers' in relation to its climate change legislation to be written into FTAs. In the resulting competitive 'race to the bottom', the high opportunity costs of diverting land, investment, and management to GHG emission reductions and sequestration embodied in conservation farming or agroecology imply that the agri-environmental 'policy reach' of ELMS will be limited [9,14]. This will be the case particularly in respect of those farms described by DEFRA as 'very large' (the top twenty-five percent of farms) and those in the 'general cropping' (cereals and horticulture) and dairy sectors in which agricultural business activities are currently profitable and enterprises are not predominantly reliant on state subsidy in the form primarily if direct payments [85–87].

The new ELMS, unless endowed with very generous budgets, will, consequently, struggle to meet such opportunity costs on the approximately fifty percent of farmed area occupied by these farm categories. Throughout much of this area, therefore, in which farmers will be preoccupied with attaining further economies of scale in the face of the discontinuation of direct payments and enhanced exposure to overseas competition, the only means by which to secure compliance with environmental objectives and GHG emissions reduction targets will be through tighter regulation. For this very reason, however, such prospective regulations are currently the subject of intense contestation [9]. Under such downward price pressures, the motivation to rely increasingly on cost-externalising imports as 'cheaps' through Surplus Extraction 2, especially, will grow, and this will be particularly significant in the dairy, poultry, and pork production sectors heavily reliant on (LUC implicated) Latin American soya imports to secure profit margins. These same pressures will also induce further farm amalgamation and the perpetuation of production capitalisation, as machinery, including new robotics, and agrochemicals are substituted for human labour. While farmers will strive to reduce energy costs and consumption where feasible through 'ecological modernisation' and 'sustainable intensification', the underlying fossil fuel intensity of production due to high capitalisation and reliance on cost-externalising imports will thwart substantive progress towards GHG reduction targets in agriculture.

Alternatively, in those zones dominated by farms described by DEFRA [85–87] as 'mixed', 'lowland grazing livestock', and 'grazing livestock LFA' (less-favoured areas, generally indicating upland landscapes), predominantly located in the west and north of the U.K. and characterised by the majority of 'small' and 'medium' farms, businesses will struggle to survive the withdrawal of direct payments [31]. The commercial activities

of farms in these categories (primarily sheep and beef cattle producers) are, on average, currently loss-making, and they remain solvent only thanks to state subsidy in the form of 'sub-hegemonic' direct payments and agri-environmental schemes [85–87]. Direct payments comprise the bulk of these payments, contributing up to one hundred percent of income in the case of many LFA farms, and their eventual withdrawal after 2028 spells the demise of many of these farms unless new ELMS subvention can make good the shortfall, an outcome that currently appears very unlikely [9] (despite the supposed 'universality' of the Sustainable Farm Incentive). This is so in part because ELMS payments are, in accordance with neoliberal and neoclassical dictates, calibrated to secure specific environmental outcomes that are deliberately decoupled from farm income considerations in order to minimise production and trade 'distortion' [81], predicated on the flawed notion that it is possible to dichotomise agricultural production from its negative or positive ecological impacts [9,12]. With future state disbursements via ELMS no longer designed to assist farm solvency, the pressures for smaller farms to sell up and for larger farms or other enterprises to absorb them will be compelling. Consequently, it seems unlikely that many of these farms will still be in existence to put into practice the Conservative government's much-vaunted ELMS. Farms in this pastoral sector have been characterised by the gradual process of extensification noted earlier [34], and their demise will lead, additionally, to the 'carbon leakage' of ruminant production to locations overseas, notably Australia and Brazil, where the GHG emission intensity of production, largely through ecologically devastating LUC, is much higher than in the U.K.

The result, overall, will be a dichotomous countryside, with agriculturally competitive (arable, general cropping, dairy, pig, and poultry) farms in the lowlands dominated by 'de-natured' market productivism and the (sheep, beef cattle) pastoral zones of the west and north reverting to 'de-socialised' 'wilderness' [56], converting to managed 're-wilding', or pursuing 'ranching-style' scale-economies. The former will intensify their dependence on the importation feed and inputs from overseas under increased competitive pressure facilitated by the FTAs currently scheduled for conclusion or negotiation with states, such as Brazil, happy to export agricultural commodities at huge ecological cost. The 'carbon leakage' of these enterprises is therefore likely to increase further. The solvency of these highly capitalised farm businesses will be thus ever more dependent on Surplus Extraction Mechanism 2, while their high GHG emissions intensity will rely, with the increasing loss of sequestration capacity in the U.K.'s intensively farmed landscapes, on the diminishing capacity degrading ecosystems overseas to continue to sequester carbon as the basis of Surplus Extraction Mechanism 3.

As for the pastoral zone of the west and north, potential gains in GHG sequestration through 'rewilding' programmes are likely to be seriously compromised by 'carbon leakage' overseas with the migration of beef and sheep production to high carbon emission locations such as Australia (sheep and beef) and Brazil (beef). 'Cheaper' beef and lamb for British consumers will be secured through the outsourcing of increased GHG emissions, with accumulation for the industry secured through Surplus Extraction Mechanism 2. In addition to failing to meet targets for GHG emission reductions in the agriculture sector, the current U.K. government's neoliberal land-use policies will, for the same reasons, also fail to meet its stated ambition for the restoration of farmland biodiversity and for the rehabilitation of soils degraded by decades of agrochemical applications. Moreover, the imperative of meeting food security needs consistent with these other objectives is wholly neglected. Consequently, questions concerning the co-production of food and biodiversity without recourse to fossil fuels whilst building sequestration capacity, together with the supply of and access to locally grown and nutritious food for all, are wholly antithetical to this radical neoliberal scenario espoused by the current U.K. government [1,9].

### 6.3. Alter-Hegemonic Discourse

Second, proponents of '**alter-hegemonic**' discourse, which perhaps may be seen in some respects to be part of the 'climate justice' narrative, advocate the putatively opposi-

tional paradigm of 'post-productivism'. This asserts that the market power of 'corporate' food interests (and here their discourse typically constructs binary between overweening corporate power and the generalised interests of 'ordinary' citizens) can be countered by exploiting the turn by consumers away from industrial food provisioning in favour of quality, organic, local, and 're-territorialised' food production [4,14]. While 'alter-hegemonic' advocates do emphasise important facets of production that are key to ecologically and climate stabilising agriculture, their paradigm, with its reliance on 'reflexive consumerism', does little to confront market dependence, capitalist relationships of production, or the imperial mode of living. Thus, the turn to so-called 'economies of scope' and niche markets and therefore to dependency on middle-class consumption as the principal revenue stream for 'post-productivist' farmers is likely to afford only temporary respite from competitive pressures as more producers enter the field of quality production with the loss of direct subsidies [4,14]. Downward pressure on prices and capital concentration are likely outcomes of this process, while pivotal dependence on higher end income for consumption of quality produce, sustained by the imperial mode of living through the main drivers of the U.K. economy premised on Surplus Extraction 1, cast considerable doubt on both the longer-term viability and the environmental/social justice claims of this 'alternative' paradigm.

This means both that producers of 'quality food' remain highly market-dependent and subject to the pressures of capitalist competition and that, where there is a significant shift to 'post-productivism' while demand for cheaper wage foods remains undiminished, productivism, as the source of those wage foods, must be undergoing extra-territorial leakage [4,14]. Such leakage, as we have seen, is typically to the periphery either in the form of finished products (for example, beef and sheep) or in the form of feed and food ingredients for further productivist elaboration in the U.K. In other words, so long as it remains subject to capitalist relations of production, 'post-productivism' implies the existence of a 'spatio-temporal' fix that externalises the costs of productivism onto the periphery. This entails 'carbon leakage', the loss of biodiversity through damaging LUC, and the thwarting of food self-sufficiency in those peripheries subject to such ecological imperialism through Surplus Extraction 2. Such 'alter-hegemony' is thus a concomitant of continued reliance on globalised food supply [14]. As such, it fails to ask the key question posed by political agroecology and food sovereignty: How can the supply of food staples for general consumption rather than merely the supply of niche markets for higher income groups be undertaken primarily within the U.K. on an ecologically sustainable, climate stabilising, and socially equitable basis? By failing to pose this question, 'alter-hegemonic' 'post-productivism' remains parasitic upon an extractive frontier of market productivism in the periphery by token of its reluctance to problematise the wider imperial mode of living, of which it comprises a key legitimating element [4]. It is a legitimating element because, rather than challenging hegemonic neoliberalism, it *complements* it as a form of *environmental* neoliberalism through its reliance on market dependence and its advocacy of neoliberally configured ELMS.

*6.4. Sub-Hegemonic Discourse*

Third, advocates of **'sub-hegemonic'** or 'political productivist' discourse [19,20,22,39,68] comprise those farming constituencies which are likely to struggle to survive with the cessation of direct payments and/or currently commercially viable enterprises that stand to suffer attenuated profit margins with increased overseas competition arising from the conclusion of new FTAs. These constituencies are represented principally by the National Farmers Union (England and Wales) and NFUS (Scotland). Both bodies adopt an 'assured income' and 'neo-mercantilist' imaginary of the future of British farming and food, harking back to the heyday of post-war national developmentalism 11,15,19], and this permeates their discursive efforts to map a 'pathway' to net-zero by 2040 [68], a more ambitious target than that set by the U.K. government. The approach of both organisations emphasises the bolstering and 'greening' of national production by means of state-backed programmes, subsidies, and high standards that, in theory, prevent 'carbon leakage' to competitors

overseas [68,88]. This discourse differs clearly from the neoliberal, 'hegemonic' vision of a transition to net-zero founded on the market rationality of individual farming entrepreneurs and sits more squarely within the 'climate justice' narrative of a technologically-driven but state-funded, 'green new deal' imaginary of the opposition Labour Party (and the Scottish National Party in Scotland).

For the NFU and NFUS (like DEFRA and the CCC), the solution to securing net-zero emissions in agriculture (and more land for biodiversity conservation) is the production of more food on less land by deploying improved yet 'sustainable' farming methods—in other words, 'sustainable intensification' or 'ecological modernisation'. This then permits more marginal land to be 'spared' for carbon sequestration through afforestation or biofuel crops or for biodiversity enhancement through 're-wilding' [68,88]. This discourse, then, as with DEFRA and the CCC, embodies a 'land sparing' rather than a 'land sharing' approach. But it is a *state-assisted*, intensified, expansionist, and technologically driven vision designed to feed the nation through capital-intensive though family-farm-based *domestic* production. The NFU and NFUS are confident, in their techno-optimism, that research will support the transition to low-carbon farming methods within an unchanged productivist configuration, with biotechnologies such as gene editing [89] being part of the innovation repertoire necessary to secure net-zero. Another facet of this, as with DEFRA's and the CCC's techno-optimism, is faith in the capacity of carbon capture and storage (CCS), with the NFU suggesting that, within its projected pathway to net-zero, the production of crops for bioenergy, carbon capture, and storage (BECCS) will be especially important, representing over half of predicted annual emissions reductions [88]. There is little hint here, however, of the considerable technical and environmental challenges and uncertainties surrounding the various forms of BECCS technologies [68]. Thus, rather than challenging productivism, it is techno-fix 'solutions' in the form, for example, of selective breeding programmes, crop genetic improvement, engineered feed additives, and slurry acidification (to reduce $CH_4$ and $N_2O$ emissions) that will facilitate the transition to net-zero. Like DEFRA and the CCC, but this time with faith not in 'markets' but rather in state-supported investment, the NFU and the NFUS assert their confidence 'in the linear onward march of technological progress and the continued evolution of the agricultural metabolization of nature through science and innovation' [68].

Tilzey [4] has, elsewhere, referred to this 'sub-hegemonic' discourse as 'neo-productivism', a form of agrarian capitalism whose rationale is the need reliably to supply mass food consumption demand in the imperium at affordable prices and supported by government interventions to foster domestic production and efficient food supply systems but with the additional safeguard of neo-imperialist actions to secure the continuing flow of 'cheap' feed, 'flex-crop' ingredients, and energy from the global South. So, while there is a greater emphasis than with neoliberalism on the production of 'finished' food goods in the imperium and commensurate state support for a wider constituency of the domestic farming community, 'neo-productivism' is still reliant centrally on the importation of cheap feed and food ingredients from the periphery to undergird the national production of 'affordable' food items for the global Northern consumer. Despite the much-vaunted claims of 'ecological modernisation' described above, neo-productivism is likely to retain much of its GHG emissions intensity at home and will, with certainty, involve considerable 'carbon leakage' through its reliance, by means of Surplus Extraction 2, on the importation of feed/food ingredients and energy 'cheaps' from the periphery. As Tilzey [4] suggests, 'neo-productivism may partially address the first contradiction of capital as under-consumption crisis in the global North, but it cannot, in the new, ecologically constrained, conjuncture, do this without encountering the second contradiction, the impacts of which will be felt differentially in the South through the new wave of extractivism'.

All three discourses described above, namely neoliberal 'hegemonic' (and 'quasi-hegemonic' environmental neoliberalism), 'alter-hegemonic' 'post-productivism', and 'sub-hegemonic' neo-productivism, thus replicate, directly or indirectly, the imperial mode of living by continued reliance on 'cheaps' extracted from the periphery (Surplus

Extraction 1 and 2), consequent carbon leakage to these zones of super-exploitation, and differential dependence on peripheral sinks to absorb GHG emissions not sequestered in the U.K. (Surplus Extraction 3).

*6.5. Counter-Hegemonic Discourse*

Finally, we may identify what Borras et al. [71] define as 'structural transformation' narratives. We suggest that these comprise '**counter-hegemonic**' responses, such as 'radical' food sovereignty [4], espousing political agroecology as a means to secure multiple ecological and social objectives in *synergy* in the form of national food self-sufficiency, with equity, climate change stabilisation, biodiversity enhancement, and soil conservation being integral components of this strategy. Proponents of 'radical' food sovereignty or political agroecology recognise the need to address the structural foundations of capitalism in order to address the climate crisis as an integral element of the ecological precarity and social inequity wrought by capitalist food systems, whether these are articulated by 'hegemonic', 'sub-hegemonic', or indeed 'alter-hegemonic' discourses. Agroecological production can meet humanity's food needs while 'cooling the planet', they argue, so long as production is designed to meet socially determined and fundamental use value needs rather than the capitalist imperative of surplus value realisation through exchange value [4,39,89]. As noted earlier, the concept of 'market dependency' is pivotal here [4,9,15,39,90]. This concept not only considers the commodification of agricultural *inputs* to be essential to capitalist or market-dependent agriculture but also the compulsion to sell *outputs* into competitive markets in order to secure the economic reproduction of the producer. Market dependency focuses on *what* is produced by farmers, asserting that when producers rely on the sale of outputs into competitive markets, even when local and 're-territorialised' as per 'alter-hegemonic' advocacy, exchange value imperatives determine not only the methods of production but also the choice of food (or indeed non-food) commodities produced and who has access to them [39]. Such market imperatives impel a preoccupation with exchange value realisation rather than the satisfaction of social needs and ecological sustainability as determined through substantive and deliberative food democracy [9,17].

**7. Outlining a Policy Framework for Agroecology, Food Sovereignty, and Degrowth in the U.K.**

*7.1. Transforming the Agri-Food System through Political Agroecology, 'Radical' Food Sovereignty, and Degrowth*

'Radical' food sovereignty advocates identify an urgent need in the U.K. for a policy framework that strongly integrates, coordinates, and synergises farming, food, environment (including climate stabilisation), health, and social equity. This represents a key element of a programme of degrowth where this is defined as 'an equitable downscaling of production and consumption that will reduce societies' throughput of energy and raw materials. . . Degrowth signifies a society with a smaller metabolism, but more importantly, a society with a metabolism which has a different structure and serves new functions' [91] (p. 3). Here, this different structure is envisaged to be necessarily non-capitalist, abjuring market-dependency, imperial reliance on surplus extraction from overseas, and reversing primitive accumulation to re-connect people equitably with the fundamental means of production, most importantly land [10]. The new functions are democratic and equitable control of essential productive resources for the satisfaction of fundamental human needs in alignment with those of more-than-human nature. A sustainable social metabolism and ecological sustainability imply that social equity and human development, as measured by the Human Development Index (HDI) of the UNDP, need to be fulfilled at far lower levels of resource consumption than are currently characteristic of the U.K. and the global North in general. This need to fulfil HDI criteria (in other words, fundamental human needs) whilst keeping resource consumption and waste deposition to a minimum has been defined by the WWF [92]. In its *Living Planet Report*, the WWF indicates that the progress of states towards 'sustainable development' (or a sustainable social metabolism) can be assessed by using the UNDP's HDI as an indicator of human wellbeing and the ecological footprint of states as a

measure of demand on the biosphere. The HDI is calculated from life expectancy, literacy and education, and per capita GDP. The UNDP considers an HDI value of more than 0.8 to be 'high human development'. Meanwhile, an ecological footprint lower than 1.8 global hectares per person, the average biocapacity available per person on the planet, could denote sustainability at the global level [92] (p. 19). Successful 'sustainable development' (sustainable social metabolism) requires that the world, on average, meets at a minimum these two criteria. Unfortunately, the global North achieves its generally high HDI only by imposing disproportionately large ecological footprint (10 global hectares in the case of the USA, only slightly less in the case of the EU and the U.K.) on the rest of the world, expressed in the imperial mode of living. A sustainable social metabolism in the U.K., as for the global North in general, would require, then, a drastic programme of degrowth through scaling back levels of resource and energy consumption and waste deposition perhaps by a factor of up to five [92].

Translating these desiderata for a sustainable social metabolism into a transformed agri-food system in the U.K. will mean, fundamentally, producing sufficient and nutritious food for all from domestic resources, importing, as a general rule, only 'non-indigenous' foods and founding such production on agroecological principles involving the cutting and, ideally, elimination of net GHG emissions, the sequestration of unavoidable GHG production by 'nature-based' means, and the conservation and enhancement of biodiversity, soils, and peatlands. The latter could be secured primarily through livestock production reduction and extensification, especially of sheep (currently numbering fifteen million in the U.K.), thereby releasing currently lost opportunities for woodland, sylvo-pastoral, and peatland conservation, expansion, and creation [93]. Concomitantly, production, distribution, and consumption of food require to be undertaken on a democratically defined basis that ensures the equitable and secure provision of healthy diets [9].[8] The basic parameters of this climate-friendly food system would comprise the elimination of grain-based meat production (for example, the cessation in the use of barley and wheat for animal feed), a proscription on the use of synthetic fertilisers and agrochemicals, the transition away from fossil fuel-based production, and the termination of imports of 'indigenous' produce and of livestock feed such as soya as part of a focus on food sovereignty through national self-sufficiency. This would, as we demonstrate below, require a significant shift from (especially grain-fed) meat diets towards vegetarianism.

### 7.2. Integrating ELMS and Agroecological Food Production

In this policy framework, ELMS payments would be integrated into support for agroecological production such that there would be co-production of food and agri-environmental benefits, including climate change mitigation. This 'land sharing' approach would be starkly different from the dichotomous 'land sparing' paradigm underpinning 'hegemonic' and 'sub-hegemonic' discourses. ELMS would therefore seamlessly align with support policy for agroecology since the two would be supporting entirely compatible rather than opposed agri-environmental and food policies. Under ELMS, within this agroecological, food sovereignty policy frame, farm management options would address three basic situations, from 'higher' to 'lower' tiers of ecological sensitivity: first, sensitive and irreplaceable sites, involving the maintenance and enhancement of semi-natural habitats; second, diversion/reversion involving the expansion and creation of semi-natural habitats; and third, agroecological production focused on the most fertile land (and least sensitive from a biodiversity perspective). Again, in stark contrast to the neoliberally configured ELMS of 'hegemonic' discourse, all farms delivering these benefits would, within a 'counter-hegemonic' policy frame, have an entitlement to an area payment, graduated according to tier, and subject to degressivity in the lowest tier for farms above a certain hectarage [9]. A strong regulatory baseline would prescribe statutory standards of land management and farming according to agroecological principles, including the proscription of synthetic fertilisers and agrochemicals.

In addition to these ELMS area payments, supporting the ecological and climate sta-bilisation dimensions of agroecology, stimulus to agroecological food production could be provided by the transformation and expansion of the current Basic Payment Scheme into a Basic Food Payment Scheme (or, more specifically, an Agroecological Area Payment Scheme)—all farms, including those under five hectares currently ineligible for BPS, would now qualify for this new payment, contingent upon an agroecological audit of the farm and accompanying recommendations for conversion to and optimal production of appropriate agroecological produce. Again, payment could be degressive for farms above a certain hectarage, although these proposals should be accompanied by a programme of land redistribution (see below). At least initially, agroecological production might receive an additional stimulus through guaranteed prices, with food then purchased by local/regional public authorities, thereby effectively severing capitalist market dependency and competi-tion. As part of this new U.K. food policy framework, the social security system should include the provision of free, healthy, and nutritious food as a basic part of the welfare package—requiring recipients, where unemployed, to participate in socially and environ-mentally useful community work and training to facilitate productive participation in the national 'green transition'. Such free provision of agroecologically produced food would also apply to state-sector schools and to the National Health Service (NHS). Elimination of the food poverty and dietary inequalities detailed earlier in this paper should be part of the ambition of any responsible government, an ambition that should be part of a comprehen-sive national plan for a 'green new deal', including transition to agroecological production. Pending the provision of decent and rewarding livelihoods for all citizens as part of this transition, the alleviation of food poverty, insecurity, and dietary inequalities should be assisted by means of public food provision.

Given the increased labour and knowledge intensity of agroecological production and conservation management [93], there will be a need for a policy of voluntary but incentivised rural re-population, and a concomitant diminution in the size of landholdings both to encourage new entrants to farming and to reflect the more 'people-' and 'nature-centred' character of agroecology. This will necessitate a policy of land reform, proscribing ownership of land above certain size limits and redistributing the resulting surplus land to new entrants to farming. As noted, land proprietorship in the U.K. is currently extremely unequal, largely a legacy of unjust and undemocratic processes of primitive accumulation implemented by landlords and larger landholders between the sixteenth and nineteenth centuries [37,97]. For reasons of social justice but, more especially, for reasons of agroecolog-ical transition through re-peopling of the countryside, this inequality in land distribution demands redress.

### 7.3. Detailing a Sustainable Social Metabolism through Agroecological Production

How might agroecological production be configured to secure the real rather than aspirational elimination of GHG emissions, carbon sequestration, ecological sustainability, and self-sufficiency in 'indigenous' food production in the U.K.? Firstly, we need urgently to eliminate all grain-based livestock rearing and to confine livestock farming to pasture-land where crops cannot be grown and that is free of synthetic fertilisers and agrochemicals. Secondly, neither conventional productivist nor 'rotational' organic production systems can generate the quantity of grain needed to supply U.K. consumption in a secure, climate stabilising, and ecologically sustainable way. This is true also of agroecological production where grain production is reliant on animal manures. This means essentially that arable and pasture must be rotated, implying, inter alia, that the potential for carbon seques-tration on what would otherwise be permanent and extensive pasture is compromised, compounded by the adverse impacts of ploughing on soil biota (see [9,93,98] for further detail). Part of Poux and Schiavo's solution is to reduce U.K. consumption of cereals by some forty-five percent, permitting, they claim, some eleven percent of the U.K. to be de-voted largely to carbon sequestration adequate to meet the U.K.'s GHG emission reduction commitments. The reduction of human grain consumption by this amount is likely to

prove very challenging, however. A potential solution is to grow cereals in a way that does not rely on animal manures as does the modelling of Poux and Schiavo [93]. Grain can in fact be grown in an agroecologically based way that increases output whilst minimising or eliminating fossil fuel usage, enhancing biodiversity, *and* sequestering carbon on a greater scale than envisaged by Poux and Schiavo. This solution addresses the two main contradictions of 'rotational' organic production—the need for high soil fertility levels, requiring rotation with livestock systems to achieve these, and the use of modern grain varieties, particularly wheat, that require these high nutrient levels and have short stems, needing frequent rotation and tillage to control weeds. These modern wheat varieties, bred to respond to fossil fuel-based synthetic fertilisers, do not grow well in low-input agroecological systems [98,99].[9]

One of the main contradictions of 'rotational' organic systems, then, is the need for ploughing or tillage. However, 'heavy and frequent tillage negatively affects a soil's physical and biological properties and is probably the most important reason for decreases in soil structural quality…Tillage may also decrease soil organic matter, which may be further reduced by rising temperatures' [102] (p. 1440). Moreover, 'minimum tillage can improve soil structure and stability, resulting in better drainage and water-holding capacity, as well as enhancing microbial activity… These practices also reduce losses of soil organic matter and thus carbon losses, while improving soil structure and water retention and enabling more permanent soil cover. There is much potential for reduced tillage to mitigate GHG emissions…' [102] (p. 1440). Other research has demonstrated how conventional tillage decreases the abundance and biomass of earthworms, with severe knock-on implications for soil structure, drainage, the recycling of organic matter into the soil, and crop production [103,104].

Tillage is thus often needed to control weeds in organic farming. Many of these biotic problems could be resolved, however, through diversification strategies such as cultivar and species mixtures to reduce infection and spread of diseases and through plant traits which confer a high level of crop competitive ability against weeds [102,105]. In addition to disease, insect, and weed control and consequently reduced or eliminated pesticide inputs, nutrient conservation, soil fertility building through increased organic matter, and enhanced yield stability are some of the ecosystem services inherently conjoined to sustainable cereal production that can be secured by crop diversification. The introduction of crop variation over time and space stabilises these systems and includes growing heterogeneous varieties that can adapt to local and changing environments, extending from the landscape to the field scale (the latter using populations or mixtures of varieties within a field). 'In systems with more variable climate and reduced external inputs, crops will need to be able to cope with spatially and temporally more heterogeneous environmental conditions. Plant breeding will have to provide varieties that are adapted to these new needs in diversified agricultural systems, which will need innovative approaches. The requirements for such varieties are enormous, as they have to combine high yield with high levels of resistance and tolerance to pests and diseases, competitiveness with weeds, and an improved stand establishment with efficient use of nutrients, water, and light. *As new characteristics are needed, breeding will have to rely on the intensive use of genetic resources (landraces, exotic, and wild resources)*' (emphasis added) [99,102,106] (p. 1441).

Many of these required traits are based on a range of genes, that is, *polygenic* inheritance, rather than on single genes, that is, *monogenic* inheritance, and are thus greatly influenced by the environment, requiring phenotypic selection. In order to produce this required polygenic inheritance, one strategy has comprised the creation of diversity through the breeding of selected varieties of modern wheat for the above desired characteristics. This technique has been pioneered by Martin Wolfe and colleagues [99]. An alternative strategy, building on the research and recommendations of Wolfe, Ostergard, and colleagues, as above [99,106], is to use landraces or 'heritage' grains, drawing on the multitude of different wheat and other cereal varieties that, until the recent past, characterised the British and European landscapes, each adapted to local soil and climatic conditions. This has the large

advantage of drawing upon *existing* (or very recently existing) polygenic inheritance, thus perpetuating or recreating agri-biodiversity, local adaptability, and resilience in the context of specific soil, biotic, and climatic conditions.

This strategy appears to be the optimum agroecological solution to the contradictions of both conventional and rotational organic cereal production. Landraces or 'heritage' grains have the great advantage of having taller stems to outcompete weeds, have higher nutritional value, and because of the need to avoid lodging (falling over due to heavy seed head in modern varieties) have lower nutrient demands than modern varieties [98,107].[10] The key here is to grow genetically diverse 'heritage' grains in the same fields, continuously, without animal manure or tillage, following a low-input approach known as continuous grain cropping (CGC) (also known as 'natural grain farming' or 'restorative continuous cropping') (see [98,107]). These cereals can be grown in this way as long as the crops are genetically diverse, have tall stems to help outcompete weeds, and all the stems are left in the field post harvest. The nitrogen removed with the grain each year is replaced by nitrogen fallout from the atmosphere, by the mineralisation of plant tissues above and below ground, and by the fixation of nitrogen by an under-sown layer of clover. Moreover, these varieties have much deeper root systems than modern varieties, enabling them to extract moisture from depth and develop, given zero tillage, complex associations with mycorrhizal fungi, greatly enhancing nutrient uptake in lower fertility soils [108]. These traits in turn confer much greater resilience in the face of weather extremes [98,107]. In short, CGC systems have the potential to greatly reduce or eliminate GHG emissions through zero application of synthetic fertiliser/agrochemicals, zero tillage, zero requirement for animal manure, and zero net oxidation of soil carbon while massively increasing sequestration through building up soil organic carbon with incorporation of cereal stems, clover, and weeds through zero tillage and through the development of carbon-rich mycorrhizal associations.[11]

CGC production yields about 2.5–3.0 t/ha even on fairly poor soils [98,107].[12] While it is necessary to note that these results, so far as it is possible to ascertain, have not as yet been published in a peer-reviewed journal despite their basis in long-term experimentation and that, therefore, further independent validation is required to confirm the long-term viability of the CGC system, they do suggest that current national demand could be met from approximately two million hectares of land (current field crop hectarage in the U.K., mostly cereals, is over six million, but a large percentage goes to feed animals and the crop land also needs to be rotated).[13] If diets were to become increasingly vegetarian/vegan, the area of CGC would need to expand further but would still be less than the current field crop hectarage. If, on a reasonable assumption, the U.K. could supply increased national demand, on the basis of increasingly vegetarian diets, from around five million hectares of land (this area of cropped land would include the agroecological production of the full range 'indigenous' crops, including those such as pulses and tubers, required for a varied and healthy human diet), this would still leave some twelve million hectares for alternative production (total farmed area in the U.K. is 17.6 million hectares [31]), including the production of extensively and agroecologically reared livestock/poultry, and greatly expanded provision for carbon sequestration. In this way, the remaining area of non-cultivated land could be devoted to extensive, grass-based livestock/dairy and other multifunctional uses involving carbon sequestration through agro-forestry and 're-wilding' [9]. The emphasis on the latter should be upon increasing the area of native woodland to considerably enhance carbon sequestration in order to meet the statutory net zero GHG ambition [32]. Under the proposal presented here, however, this contribution to net zero does not exclude the considerable contribution to GHG reduction and sequestration performed by areas devoted primarily to food production, both 'infield' through techniques such as CGC and also 'field edge', through the contribution of natural features such as hedgerows [9]. This is the essence of a land sharing approach, seeking to secure sustainability across the landscape, even if there are differences in emphasis between areas of high soil fertility on the one hand and areas of agriculturally marginal land more suited to 're-wilding' on the other.

Such a programme of 'counter-hegemony' through 'radical' food sovereignty, agroecology, and degrowth would, by securing net-zero GHG emissions, carbon sequestration, and food self-sufficiency, effectively eliminate, at least in the agriculture sector, the U.K.'s reliance upon the three surplus extractive mechanisms detailed earlier. It would, in short, eliminate the U.K.'s parasitic imperial mode of living. This would, then, enable the U.K. to secure effective self-sufficiency in indigenous food whilst simultaneously securing food equity, landscape-scale biodiversity, and biophysical resource conservation through agroecological land sharing and the realistic and expeditious rather than merely aspirational attainment of the net-zero GHG emission target. In short, this would enable the U.K. to secure a sustainable social metabolism, at least so far as the agri-food system is concerned.

*7.4. Politico-Economic and Ideological Constraints Imposed by the 'Imperial Mode of Living'*

Sadly, however, political awareness of, and political commitment to, such a programme of radical but necessary and *feasible* (not merely rhetorical) transformation is virtually non-existent in the U.K. largely, and ironically, due to the operation of the imperial mode of living itself. Thus, within the U.K. farming community, for example, such a programme of radical transformation would be likely to find support currently only amongst a very few interest groups such as the Land Workers' Alliance, affiliated to *La Via Campesina*, membership of which is numbered only in the hundreds. The unfortunate reality appears to be that the legitimacy and material basis of the imperial mode of living will likely require to be compromised *before* such an agroecological transition can occur. In other words, the necessary agroecological transition would seem to require the prior fracturing or at least severe attenuation of the imperial mode of living before it receives the necessary groundswell of political support. Were the contradictions of neoliberalism to continue to erode the livelihoods of the small and medium farm constituencies, compounded by the acceleration of the climate emergency, such current supporters of sub-hegemony and particularly of alter-hegemony could perhaps be persuaded of the merits of a political agroecological transition, with the proviso that such a transition were adequately funded. The greatest opposition seems likely to derive from the larger farm constituency and, given the pervasiveness of 'propertisation' [114], especially directed to any notion of land redistribution. Indeed, if developments in Germany, the Netherlands, and elsewhere are any guide, threats to absolute property rights, whether such rights are real or 'phantom' [114], appear to be leading to a shift in political affiliation to the right. The popularity of Brexit amongst the farming community (at least initially until its radical neoliberal intent became clearer) suggests that this rightward shift is also present in the U.K.

While the mounting impacts of austerity and now climate change are beginning, belatedly, to fracture neoliberal hegemony amongst the wider population, especially amongst a younger demographic, it seems likely that the bulk of popular opposition will be directed and co-opted into various forms of sub-hegemony, seeking to restore incomes, consumerism, and growth through a slightly more interventionist form of 'green' and 'redistributive' capitalism. While this may alleviate a degree of poverty amongst those who have suffered under austerity and make incipient moves to reduce fossil carbon dependency, it does very little to set the U.K. on a path of real sustainability, let alone of degrowth. Even here, the reluctance of the prospective new incumbents of government (The Labour Party) appear very unwilling to pursue any agenda that could be construed as anything more than mildly reformist. More likely, for the time being, then, is maintenance of something approximating the 'hegemonic' status quo in the field of agri-food, climate change, and ecological sustainability—through 'techno-fix' programmes of 'ecological modernisation', supplemented by 'creative carbon accounting'—than the likelihood of meeting the net-zero target recedes, whereby green-washed 'spatio-temporal' fixes are deployed, deferring the necessary actions and costs that the U.K. needs to take or absorb. For the farmed environment, this will likely involve a continued focus on 'infield' 'techno-fix' programmes, accompanied by 'field edge' or marginal land environmental initiatives supported by ELMS, perpetuating the dominant land sparing approach in which infield productivism

is juxtaposed to 're-wilding'. This will serve to thwart the imperative for an integrated land sharing approach, one which synthesises the need to address simultaneously climate change, ecological sustainability, and food insecurity through a political agroecological and food sovereign transition.

## 8. Conclusions

By reference to empirical indices, this paper has laid out at length the essential unsustainability of the U.K. food system along the dimensions of climate change, biodiversity loss and decline, and food (in)security. It has also identified the causality underlying these indices of unsustainability, pointing to the prevalence of capitalism and especially neoliberalism in generating the multiple contradictions of the U.K. food system, including its adverse impacts on countries overseas through reliance on cost-externalising 'cheap' global supply. The paper has also shown how dominant 'hegemonic' and 'sub-hegemonic' politico-economic interests and their accompanying discourses have both generated these contradictions and shaped mitigatory responses to them, predominantly as symptom management. Since both are wedded to productivism and continued economic growth, it seems improbable, despite attempts—real or rhetorical—to decarbonise continued capital accumulation, that any real strides will be made in securing *integrated* solutions to the social and ecological contradictions of the U.K. food system. Rather, responses will be piecemeal, mitigatory (symptom management), and reliant on continued cost-externalisation overseas. Oppositional discourses are available, however. The first, a discourse we have nominated as 'alter-hegemony', strongly advocates a land sharing approach by integrating farming, biodiversity, carbon sequestration, and localised production-consumption relations at the territorial level—a form of bioregionalism [1]. As such, however, the problem is seen to be primarily one of scale rather than an issue of capitalist relations of production—inequalities and private social–property relationships are as entrenched at the local level as much as they are nationally and internationally. Failure to address social inequity 'locally' will lead to the persistence of food poverty and dietary inequalities, and ignoring the enduring presence of capitalist social property relationships will generate unavoidable pressures for continued growth and competition. It is salutary to recall that capitalism began 'locally' within the context of unequal class relationships (see [18]).

Given these shortcomings of 'alter-hegemony', we have argued that a second oppositional discourse, 'counter-hegemony', alone offers an integrated approach to simultaneously resolving the problems of food (in)security and social inequity on the one hand and ecological sustainability (subsuming climate change stabilisation) on the other. Such 'counter-hegemony', embodied in 'radical' food sovereignty, political agroecology, and degrowth, proposes the abrogation of capitalist social–property relationships. This entails the supersession of abstract capitalist market dependency (the rule of the market as an impersonal force) by means of concrete democratically (politically) determined systems of localised governance, overseeing equality of access to the means of food production and to the fruits of that production. This requires a rediscovery of political agency, solidarity, mutuality, and ways of nurturing our humanity by respecting non-human nature.

**Funding:** Research for sections of this paper was supported by Research England QR Strategic Priorities Fund.

**Data Availability Statement:** The raw data supporting the conclusions of this article will be made available by the authors on request. Data is publicly available due to privacy reason.

**Conflicts of Interest:** The author declares no conflicts of interest in this paper.

## Notes

[1]    By capitalist here we also mean market-dependent family farms, even though these may not employ off-farm labour. The peasantry (self-subsistent and semi-self-subsistent agrarian producers) had effectively disappeared from Britain by the mid-19th century (see [4] and [18] for more detail on the rise of agrarian capitalism in Britain).

2     Appropriationism and substitutionism refer to the undermining of discrete element of the agricultural production process, their transformation into industrial activities, and their re-incorporation into agriculture as inputs, for example, human labour by machinery, animal traction by the tractor, manure by synthetic fertilisers, etc. [23].

3     The 'imperial mode of living' refers to the normalisation of affluence, growth, and high levels of resource consumption characteristic of the global North (the imperium), predicated significantly upon the ideologically 'invisible' exploitation of the global South.

4     The authors (WWF and RSPB) assigned a risk score to each U.K. sourcing country based on their deforestation/conversion rates, labour rights, and rule of law indices. Scores varied from 0–12, with 11 or above being 'very high' risk and 9–10 being 'high' risk.

5     The uncompensated appropriation of land and resources by capital for wealth accumulation, involving the wholesale removal of the original inhabitants without absorption as labour into the subsequent agro-industrial or mining developments.

6     Key farmland and ground-nesting bird species, such as skylark, lapwing, and stone curlew, require no or very low vegetation when incubating eggs in order to see and avoid predators—autumn sown cereals are already too high in early spring to enable these species to incubate safely. Autumn-sown cereals are bred to respond to synthetic fertilisers and put on growth very quickly; traditional or 'landrace' cereals, even when germinating in the autumn, do not produce significant growth until the next spring, especially when grown in organic and no-till management systems—they may produce less per area than modern cereals grown with agrochemicals, but they can produce indefinitely and sustainably with no artificial inputs and generate no negative ecological externalities.

7     Despite the fine words expressed in DEFRA's Agricultural Transition Plan update of January 2024 and the improved payment offers and increased coverage/flexibility of the ELM schemes detailed therein, the essential principles of land sparing 'public goods' payments, embodying a dichotomy between productivist farming on the one hand and biodiversity conservation on the other, remain in place.

8     The characteristics of a diet consistent with public health, climate change stabilisation, and low environmental impact are already quite clear [94–96]. This is a diet that provides diversity, with a wide variety of foods consumed; achieves balance between energy intake and energy needs; is centred around minimally processed whole grains, tubers, and legumes, fruits, and vegetables; has moderate/small amounts of meat, dairy, unsalted seeds and nuts; has small quantities of fish from certified fisheries; has oils and fats with a beneficial omega 3:6 ratio such as rapeseed and olive oil; and is very limited consumption of foods high in fat, salt, and sugar and low in micronutrients.

9     A recent paper [100] appears at first sight to contradict this statement. Closer examination, however, shows this not to be the case. The study on which the paper is based only tests varying levels of agrochemical inputs on modern wheat varieties, with the 'lowest input' still at $110\,\mathrm{kgNha}^{-1}$. This, however, is not 'low input' from an agroecological perspective, where the expectation is that *no* agrochemicals (or, more specifically, synthetic fertilisers) are employed. The modern varieties tested in this study would certainly not thrive under a zero-agrochemical regime. Moreover, the application of N at the lowest rates in the study would still prove toxic to most non-target plant species in the field or field edge (the great majority or wild plant species find even very low levels of N application toxic [101]), and this applies also to the soil biome—agroecology seeks to maximise the vigour of this soil biome by refraining from agrochemical use to support sustainable, resilient soils and hence sustainable and resilient cultivar production.

10     Although the work of John Letts as an academic archaeobotanist has been widely published in peer-reviewed journals, his long-standing experimental work with 'heritage' grains and CGC has, so far as it is possible to ascertain, not yet been similarly published (although it has been published in non-peer reviewed publications as per the 'Land' citation in the present paper). However, the agroecological foundations for his fieldwork and conclusions from it are supported by peer-reviewed research (see above), and his work has been funded through the EU Horizon 2020 Research and Innovation Programme under Grant Agreement No. 727848 and is summarised in the following link entitled 'Low input and organic heritage cereal production in South East England': http://cerere2020.eu/wp-content/uploads/2020/03/17_EN.pdf (accessed on 15 December 2023). Similar experimental fieldwork and findings have been undertaken and generated in the USA by Rogosa (funded by the USDA Sustainable Agriculture Research and Education Program), where einkorn, emmer, and other landrace wheats outperform modern wheats under organic conditions (that is, where synthetic fertilisers and pesticides are not applied) [107] (p.4).

11     It may be asked how CGC and agroecology are connected to related (or putatively related) production techniques such as conservation agriculture (CA) and circular agronomy. Concerning CA, this, according to the FAO's definition [109], is a farming system that can prevent losses of arable land while regenerating degraded lands. It promotes the maintenance of a permanent soil cover, minimum soil disturbance, and diversification of plant species. It enhances biodiversity and natural biological processes above and below the ground surface that contribute to increased water and nutrient-use efficiency and to improved and sustained crop production. CA principles are universally applicable to all agricultural landscapes and land uses with locally adapted practices. Soil interventions such as mechanical soil disturbance are reduced to an absolute minimum or avoided, and external inputs such as agrochemicals and plant nutrients of mineral or organic origin are applied optimally and in ways and quantities that do not interfere with or disrupt the biological processes. This definition is virtually identical to CGC and agroecology with the exception that these avoid agrochemicals altogether since they recognise the damage that agrochemicals cause to soil, soil biota, and non-target field and field edge plant species, thus compromising the underlying rationale of conservation agriculture itself.'Circular agronomy', for its part, aims to close nutrient cycles in the agri-food chain, aiming to improve the current carbon,

nitrogen, and phosphorus cycling in agro-ecosystems and related up- and downstream processes within the value chain of food production [110]. This, however, seems to be part of an 'ecological modernisation' agenda tied to capitalist productivism. Agroecology is based centrally on such circularity, of course, but without recourse to synthetic fertilisers or mineral supplements that generate major problems for the environment, soil health, and the longer-term sustainability of food production itself.

[12] It may appear that these figures are contradicted by the long-term wheat yield trials at Rothamsted Experimental Station in the U.K. These trials show a yield of about 1 tonne/ha under continuous wheat cropping and 2 tonnes/ha when wheat is grown in rotation [111]. The results of CGC and Rothamsted are not directly comparable, however. This is because (a) the CGC method is no till, while the Rothamsted plots are ploughed annually; (b) CGC does rely (in part) on chopped straw and clover to retain fertility levels, so this is not directly comparable to the continuous cropping without fertiliser undertaken at Rothamsted. In other words, the 1 tonne/ha yield at Rothamsted is based on continuous cropping of wheat without any fertiliser application, which is not the same as CGC. A more meaningful comparison with CGC would be the continuous cropping with farmyard manure (FYM) application trial at Rothamsted, which demonstrates yields between 2 and 3 tonnes/ha for most of the experimental period (rising up to 6tonnes/ha after 1970 with change in wheat variety). But, as pointed out above, FYM relies on livestock which means diverting considerable areas of land to livestock production to retain the fertility of cropped areas.

[13] In fact, 15 million tonnes of wheat are produced annually in the U.K., but only c. 5 million tonnes are milled to produce flour for human consumption—two-thirds of wheat produced is fed to animals. Animal feed grains are not suitable for flour milling, however. As argued above, all cereal production should be directed to human, not to livestock, consumption. However, this cereal should be produced on an agroecological basis without recourse to agrochemicals, synthetic fertilisers, or to livestock to provide the FYM for organic rotations. As argued above, this shift to non-rotational agroecological production is both necessary and feasible. In addition to the multiple disbenefits of conventional wheat production identified above, it also needs to be pointed out that modern varieties of wheat and conventionally milled wheat flour (through the Chorleywood method), together with the standard addition of sugar and other additives to bread so manufactured, has important negative health and nutritional impacts [112,113].

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
