# Peer review of "Ill Fares the Land: Confronting Unsustainability in the U.K. Food System through Political Agroecology and Degrowth"

_land, doi:10.3390/land13050594_

Round 1
Reviewer 1 Report
Comments and Suggestions for Authors
Greetings! This manuscript is an overview study. It highlights the unsustainable characteristics of the UK food system by summarizing previous research to display the links between the UK food system and some of the key sustainability indicators. This manuscript has important implications for subsequent research into the sustainability of the UK food system. In my opinion, there are three aspects that need to be improved.
(1) Words need to be condensed.
(2) Add the subheadings (2.1,2.2,2.3….) to each section makes for a more organized presentation.
(3) Add charts and graphs to present article content.
Author Response
Dear Reviewer,
Thank you for your comments. I have added sub-headings as per your helpful suggestion. I am also happy to add figures to the text in the final version, if that is deemed necessary. Reviewer 4 supplied extremely supportive and detailed comments on the manuscript, indicating acceptance of the text essentially without change. In view of these comments from Reviewer 4, I am therefore very reluctant to tamper with the integrity of the text, something that would be required in order to condense it as per your suggestion. I trust you will understand this reluctance on my part and my desire to retain the integrity of the text as it stands.

Reviewer 2 Report
Comments and Suggestions for Authors
Dear Author,
this study is an important eye opener and highly relevant. However, some of the proposed solutions are not fact-based. The paper relies heavily on opinions stated in 2 reports:
99. Mason P and Lang T (2017) Sustainable Diets: How Ecological Nutrition Can Transform Consumption and the Food System. 1887
London: Routledge. 1888
101. Letts J (2020) Continuous Grain Cropping. Land 27: 28-34. https://www.thelandmagazine.org.uk/articles/continuous-grain-cropping
Please refrain from using these 2 dubious reports and only cite papers that are based on field experiments.
I am listing a few examples that need changes:
L 633: “Shift from spring-sown to autumn-sown cereals leading to loss of nesting sites for characteristic farmland bird species;”
Any reference for this? Please consider that not using the land to grow food during the winter months would lead to a decline in production. Spring cereals yield less than winter cereals, because the growth duration is shorter.
L 1549 “These modern wheat varieties, bred to respond to fossil fuel-based synthetic fertilizers, do not grow well in low input, agroecological systems [99,100].”
This contradicts the findings of a large study:
https://www.cimmyt.org/news/modern-wheat-breeding-benefits-high-and-low-input-farmers-study-shows/
https://www.nature.com/articles/s41477-019-0445-5.epdf?shared_access_token=xAQrQ1zS57u_7UA0QRA-6tRgN0jAjWel9jnR3ZoTv0OjW7AKTN4iRXe5CwSatCcf4WecHVXvVlaUvt2xMn4pBbPPJB8qVjUmLWppBsx2TI5tPJdbYd9E9caFqFWwnpHRTl7V5jaMCALHHKIGt5VrSQ%3D%3D
L 1551] “The agroecological solution to these contradictions is to use landraces, or ‘heritage’ grains, drawing on the thousands of different wheat, and other cereal, varieties that used to characterize the British landscape, each adapted to local soil and climatic conditions. These have the great advantage of having taller stems to outcompete weeds, have higher nutritional value and, because of the need to avoid lodging (falling over due to heavy seed head in modern varieties), have lower nutrient demands than modern varieties [99,101].”
[99 and 101] are not based on research. These are just statements made by 2 authors, without evidence.”
There are other solutions, such as Circular agronomy and conservation agriculture.
[L 1574] CGC production yields about 2.5-3.0 t/ha even on fairly poor soils [99,101].
The Rothamsted long-term trials show a yield of about 1 t/ha under continuous wheat (Fig 1) and 2 t/ha (Fig 2) when grown in rotation:
https://www.rothamsted.ac.uk/sites/default/files/national-capability/long-term-experiments/Web_LTE%20Guidebook_2019%20Final2.pdf
There are other solutions: Only about 1/3 of UK wheat production is used for direct human consumption. According to
https://magazinebbm.com/blog/flour-and-bread-market-in-the-uk-1473
the production is 15 million tonnes. Only about 5 million tonnes are milled. Same goes for the EU: only about 20% of its wheat production is used for direct human consumption.
Author Response
Dear Reviewer,
Thank you for your helpful and incisive comments and your suggestions for possible changes to the text. I believe I have addressed all the points you raise through additions/changes to the text (these additions/changes are identified as track changes). I believe I have responded meaningfully to all the points you raise through these additions/changes to the text. I have argued in favour of retaining the references that you suggest I refrain from using, since these are in reality based on long-standing research even if they appear not as yet to have been published in relevant peer-reviewed journals (I believe you may have misidentified one of the references in question - Mason and Lang should be Rogosa?). I hope you find that my additions/changes to the text address satisfactorily all the points that you raise.

Reviewer 3 Report
Comments and Suggestions for Authors
The paper presents very important topics and it is very important for the current public date but... it is almost impossible to read. In my opinion the paper should be divided in at least two papers - one on the UK food system and the other on the agriculture and degrowth.
I am not sure if the title starts with a mistake or a Romanian 3 - which is correct?
Author Response
Dear Reviewer,
Thank you for your comments. For your information, the title does not comprise Roman numerals but is ILL Fares the Land, an extract from Oliver Goldsmith's well-known 18th century poem.
I am grateful for your opinions on my paper. Unfortunately, however, your views appear to be diametrically opposed to those of Reviewer 4, which makes it rather difficult for me to respond substantively to them. Reviewer 4, who responded with substantive commentary, found my manuscript 'a pleasure to read'. Based on Reviewer 4's detailed and very supportive commentary, it would seem that to undertake any significant change in language would be to go against these comments. I feel fully justified, therefore, in leaving the paper more or less as it is (except for some additions and very small changes in response to Reviewer 2). Regarding the suggestion that the paper be split up - the purpose of the paper is to provide a comprehensive assessment and critique of the UK food system (subsuming agriculture) from the perspective of political agroecology and degrowth. There is, therefore, an inherent dialectic between the data presented and the critique throughout the piece: to split it up would be to lose this integrated approach and the relationship between empirical evidence and my theoretical critique of it. Again, the extremely supportive comments of Reviewer 4 surely argue strongly against splitting up the paper. I hope this affords adequate justification for retaining the paper as one integral piece of work and I hope very much you will understand my motivation here. Thank you again for your comments.

Reviewer 4 Report
Comments and Suggestions for Authors
This article was a pleasure to read. I learned a lot. It reviews a significant recent literature describing the fundamental ecological and social contradictions of the UK food system, and goes further to identify what is missing in this literature: a description of the causal mechanisms in productivist, capitalist, and neoliberal agriculture, which has hegemonic status. It shows how the current system displaces the harms of this system onto poor countries in an imperial fashion through "carbon leakage" and massive unsustainable land use transformations, as well as onto the diets of poorer communities in the UK. The article is a welcome antidote to the tecno-optimistic discourse promoted by agribusiness and Conservatives that minimize structural problems and prevent adequate policy responses. The article sounds the alarm by showing how market oriented solutions divert urgent critiques of capitalism in light of unfolding ecological calamity, preventing the significant land use changes necessary, and instead arguing that the lack of capitalist markets is the culprit. This is important political economic analysis. In addition, the article criticizes shallow alternative discourses ("alter hegemonic" post-productivism), which remain reliant on the market model and constitute a form of environmental neoliberalism, and compares them to the more robust transformative proposals of food sovereignty and political agroecology, identified as "counterhegemonic" discourses. The article does important work by connecting these proposals to the degrowth arguments and identifying policy changes necessary to bring these alternatives to scale, such as linking Environmental Land Use payments to agroecology, a land sharing rather than a land sparing approach to conservation, a transformation and expansion of the Basic Payment Scheme to include agroecologically produced food, accompanied by land redistribution, price supports for agroecology, integration of agroecology into social security, voluntary rural repopulation (repeasantization), curtail commercial livestock, eliminate grain based livestock production, shifting to landrace and heritage cereal grains. These are radical solutions, but the article argues that they are certainly feasible, but incompatible with the imperial mode of living--a tall order that the article recognizes. But it is important to speak the truth, however unpopular, as a first step to building coalitions for the kind of transformation that is so urgently necessary to address the problems that we have either been led to believe will be solved by market mechanisms (they will not), or cannot be solved at all.
Author Response
Dear Reviewer,
Thank you so much for your extremely supportive comments. This serves as a strong vindication of my arguments and of the empirical material deployed to support them. In response to the other reviewers, I have made some additions and small changes to the text, overwhelmingly in the final section concerning a policy framework for political agroecology and degrowth. These serve, I hope, to further substantiate my arguments rather than to change any of them.
Thank you again for your very supportive review.

Round 2
Reviewer 2 Report
Comments and Suggestions for Authors
The author has addressed some of my concerns, but the paper still tends to condemn certain practices and generalize and extrapolate from poorly documented literature.
Example:
L 630 to 643: The paragraph starts with “The environmental impacts of productivism can be enumerated as a series of generic issues:
On L 643 it states: Shift from spring-sown to autumn-sown cereals leading to loss of nesting sites for characteristic farmland bird species”
The cited report by John Letts (2020) on “Continuous Grain Cropping” states: The success of this approach depends on six principles:
(i) early autumn planting of winter cereals that require vernalisation (ie a cold period).
Question: does John Letts belong to the camp of “Productivists”?
One cannot just cherry-pick and idealize the positive aspects of certain technologies while leaving out other aspects.
Letts J (2020) Continuous Grain Cropping. Land 27: 28-34. https://www.thelandmagazine.org.uk/articles/continuous-grain-cropping
+++++++++++++
L 1647: You wrote“….suggest that current national demand could be met from approximately two million hectares of land (current field crop hectarage in the UK, mostly cereals, is over six million, but a large percentage goes to feed animals and the crop land also needs to be rotated).”
People need a diverse diet. Cereals alone cannot feed the people of the UK. Hence, CGC does not solve all problems. Some crop land (and plant-nutrients) will be required to grow potatoes, pulse crops, sugar beet, etc.
In my personal opinion, the rhetoric still needs to be toned down. It might be worthwhile to emphasize that these alternative production systems require more research.
Author Response
Dear Reviewer
Thank you for your further comments on my manuscript. I address your comments point-by-point below:
Reviewer Comment: The author has addressed some of my concerns, but the paper still tends to condemn certain practices and generalize and extrapolate from poorly documented literature.
Author Response: I believe my criticism of practices associated with fossil-fuel and agrochemical-based productivism is well founded. My paper documents at length the negative climate change, land use footprint, and biodiversity impacts of such productivism. In laying out the merits of an agroecological approach, I do build on well researched and documented literature, cited in the paper (for example, Wolfe et al. and subsequent references). The work of Letts, in turn, builds on this empirically grounded and documented work and I explain why I suggest his work is especially well-founded on many years of field experiments despite its lack of presence in peer-reviewed journals. I do make the point that Letts' research does require further independent validation to confirm the long-term viability of the CGC system (Lines 1627-1631). However, wider agroeocological research, empirically-based and independently validated, constitutes a very strong foundation from which to generalise critiques of fossil fuel and agrochemical-based productivism.
Reviewer Comment:
Example:
L 630 to 643: The paragraph starts with “The environmental impacts of productivism can be enumerated as a series of generic issues:
On L 643 it states: Shift from spring-sown to autumn-sown cereals leading to loss of nesting sites for characteristic farmland bird species”
The cited report by John Letts (2020) on “Continuous Grain Cropping” states: The success of this approach depends on six principles:
(i) early autumn planting of winter cereals that require vernalisation (ie a cold period).
Question: does John Letts belong to the camp of “Productivists”?
One cannot just cherry-pick and idealize the positive aspects of certain technologies while leaving out other aspects.
Author Response: This comment does not contradict my argument. I make the point in Footnote 6 that: 'traditional or 'landrace' cereals, even when germinating in the autumn, do not produce significant growth until the next spring, especially when grown in organic and no-till management systems'. Letts emphasises the point that early autumn planted winter cereals require vernalisation either before they germinate or before they put on significant growth. This means that, unlike modern autumn-sown cereals, 'landrace' cereals, even when autumn-sown, do not put on enough growth in the early spring to compromise the breeding of ground-nesting bird species. In conclusion, John Letts does not, therefore, belong in the camp of the 'Productivsts'; and I am not cherry-picking and idealising selected features of CGC or agroecology.
Reviewer Comment:L 1647: You wrote“….suggest that current national demand could be met from approximately two million hectares of land (current field crop hectarage in the UK, mostly cereals, is over six million, but a large percentage goes to feed animals and the crop land also needs to be rotated).”
People need a diverse diet. Cereals alone cannot feed the people of the UK. Hence, CGC does not solve all problems. Some crop land (and plant-nutrients) will be required to grow potatoes, pulse crops, sugar beet, etc.
Author Response: I am not suggesting that diet should be restricted to cereals. I am merely suggesting that the cereals component of diet can be produced by means of CGC, given that cereals will no longer be fed to livestock. As I point out, livestock can still be reared on grass in areas not suitable for crop production, preferably as part of agroforestry production systems, these areas producing fruit and nuts in addition to meat and dairy (this represents some two-thirds of the UAA in the UK). The area envisioned to be devoted to crop production (c. 5 million ha) will not be used just for cereal production, although this, as at present, will form the majority of this area, but will be used to grow the full diversity of 'indigenous' crops required for a healthy human diet, including pulses and potatoes. If this is not sufficiently clear in the manuscript, I have taken the precaution of adding a note of clarification on page 34 (see track change). As I note, without the need to feed animals from grain or use them for rotation, the bulk of current croppable area (6.1 million ha, which includes non-cereal crops) could be used to feed humans on the basis of largely (but not necessarily exclusively) vegetarian (and climate friendly) diets - this, as indicated above, includes the production of non-cereal crops grown in an agroecological manner (although experimental work is ongoing, potatoes, for example, can be grown on the basis of no-till, something that is urgently required given the current adverse impacts of productivist potato cultivation). The area of sugar beet production could be substantially reduced in line with the need to reduce sugar in diets.
Reviewer Comment: In my personal opinion, the rhetoric still needs to be toned down. It might be worthwhile to emphasize that these alternative production systems require more research.
Author Response: I do not accept the accusation that my paper uses rhetoric. My critique of productivism is founded in detailed empirical documentation of its multiple adverse impacts, and my policy recommendations and conclusions derive from this detailed documentation. Where appropriate, I have indicated that alternative production systems may require more research and documentation (see above).
Thank you again for your comments. I hope I have addressed satisfactorily all the issues you raise.
Reviewer 3 Report
Comments and Suggestions for Authors
I my opinion the paper is very interesting but covers too many issues in too much detail. There probably should be a limit to the lenght of the paper.
In my initial review I suggested dividing the paper into 2-3 more focused papers. The author decided to make the paper even longer. In my opinion thus makes the paper not improved but the author mentioned that a different reviewer made opposed suggestions.
Author Response
Dear Reviewer
Thank you for your further comments. Please see my point-by-point response below.
Reviewer Comment:
I my opinion the paper is very interesting but covers too many issues in too much detail. There probably should be a limit to the lenght of the paper.
In my initial review I suggested dividing the paper into 2-3 more focused papers. The author decided to make the paper even longer. In my opinion thus makes the paper not improved but the author mentioned that a different reviewer made opposed suggestions.
Author Response: I am grateful for your opinions on my paper. Again, I can really only repeat my response to your first round comments. Thus, unfortunately your views appear to be diametrically opposed to those of Reviewer 4, which makes it rather difficult for me to respond substantively to them. Reviewer 4, who responded with substantive commentary, found my manuscript 'a pleasure to read'. Based on Reviewer 4's detailed and very supportive commentary, it would seem that to undertake any significant change in language would be to go against these comments. I feel fully justified, therefore, in leaving the paper more or less as it is (except for some additions and very small changes in response to Reviewer 2). Regarding the suggestion that the paper be split up - the purpose of the paper is to provide a comprehensive assessment and critique of the UK food system (subsuming agriculture) from the perspective of political agroecology and degrowth. There is, therefore, an inherent dialectic between the data presented and the critique throughout the piece: to split it up would be to lose this integrated approach and the relationship between empirical evidence and my theoretical critique of it. Again, the extremely supportive comments of Reviewer 4 surely argue strongly against splitting up the paper. I hope this affords adequate justification for retaining the paper as one integral piece of work and I hope very much you will understand my motivation here. The paper is now slightly longer than the original version due to the need to respond in some detail to the comments of Reviewer 2 - this, I'm afraid is unavoidable.
I hope you can understand my desire and need (given other reviewers supportive comments) to retain the paper intact. Thank you again for your comments.